# Assessing the relationship between early maladaptive schemas and interpersonal problems using interpersonal scenarios depicting rejection

**Thomas Janovsky**[1]*, **Adam J. Rock**[1], **Einar B. Thorsteinsson**[1], **Gavin I. Clark**[2], **Valerie Polad**[3], **Suzanne Cosh**[1]

1 School of Psychology, University of New England, Armidale, New South Wales, Australia, 2 School of Psychology, Newcastle University, Newcastle upon Tyne, Tyne and Wear, England, United Kingdom, 3 Department of Psychology and Neuroscience, University of North Carolina at Chapel Hill, Chapel Hill, North Carolina, United States of America

* tjanovs3@une.edu.au

**Data Availability Statement:** All relevant data are within the paper and its supporting information files.

## Abstract

### Background

Early maladaptive schemas (EMSs) have been theorised to contribute to reoccurring interpersonal problems. This study developed a novel experimental paradigm that aimed to assess if EMSs moderate the impact of interpersonal situations on interpersonal responses by manipulating the degree of rejection in a series of interpersonal vignettes depicting acceptance, ambiguous rejection and rejection.

### Method

In a sample of 158 first-year psychology students (27.2% male; 72.2% female; 0.6% other) participant responses to interpersonal scenarios were measured including degree of perceived rejection, emotional distress, conviction in varying cognitive appraisals consistent with attribution theory and behavioural responses to scenarios. Qualitative data was analysed using inductive content analysis and statistical analyses were conducted using multilevel mixed effect linear and logistic regression models using the software Jamovi.

### Results

People reporting higher EMSs reported increased emotional distress ($F(1, 156) = 24.85$, $p < .001$), perceptions of rejection ($F(1, 156) = 34.33$, $p < .001$), self-blame ($F(1, 156) = 53.25$, $p < .001$), other-blame ($F(1, 156) = 13.16$, $p < .001$) and more intentional ($F(1, 156) = 9.24$, $p = .003$), stable ($F(1, 156) = 25.22$, $p < .001$) and global ($F(1, 156) = 19.55$, $p < .001$) attributions but no differences in reported behavioural responses. The results also supported that EMSs moderate the relationship between interpersonal rejection and perceptions of rejection ($F(2, 1252) = 18.43$, $p < .001$), emotional distress ($F(2, 1252) = 12.64$, $p < .001$) and self-blame ($F(2, 1252) = 14.00$, $p < .001$).

**Funding:** This research has been conducted with the support of the Australian Government Research Training Program Scholarship.

**Competing interests:** The authors have declared that no competing interests exist.

## Conclusion

Together these findings suggest that people with EMSs experience increased distress and select negative cognitions in situations where there are higher levels of rejection but that distress and negative cognitions are generally higher in people with EMSs irrespective of the situation.

## Introduction

### EMS and interpersonal problems

Interpersonal problems are a form of psychological distress associated with our interpersonal relationships with an individual or group of individuals [1]. Interpersonal problems are among the most common reasons people seek psychological support and are often reported in clinical populations as well as being key diagnostic markers of personality disorders [2, 3]. Recurrent interpersonal problems also contribute to poorer treatment outcomes in therapy including increased dropout rates, increased disordered symptoms and lower remission rates [4, 5]. Therefore, understanding how recurrent interpersonal problems are caused and maintained is essential for guiding treatment and improving outcomes for people who experience difficulties with mental health.

Researchers have long argued that recurring interpersonal problems may arise from "schemas" or maladaptive beliefs about ourselves, others and the world [6–8]. Early maladaptive schemas (EMSs) form the foundation of Schema Therapy, which is currently one of the most comprehensive theories that describes the role of schemas in psychopathology [7]. It is theorised that EMSs arise from unmet psychological needs during childhood such as needs for autonomy, safety or connection [7]. Young, Klosko and Weishaar [7] argue that EMSs go on to form the building blocks of our personality, guiding how we think, feel and behave in response to a range of situations in the future.

To date, there have been a large number of studies supporting a link between EMSs and interpersonal problems [e.g. 9–16]. People who report higher levels of EMSs have been found to report higher levels of aggression [14, 15], domestic violence victimisation [16], interpersonal conflict [17], marital and peer relationship dissatisfaction [18, 19], bullying victimisation [20], and unhelpful interpersonal behaviour such as avoidant, vindictive, controlling or submissive patterns of behaviour [1, 21]. A recent meta-analysis found a moderately strong relationship between EMSs and interpersonal problems, and EMSs related to beliefs about disconnection and rejection had stronger associations with interpersonal problems when compared to other EMSs [22]. Together, these studies provide support for an association between EMSs and interpersonal problems, particularly for schema related to rejection.

### EMSs and experimental designs

Researchers still do not fully understand how EMSs are associated with interpersonal problems. Most of the research investigating EMSs and interpersonal problems has been limited to cross-sectional studies [22]. It is theorised that people with EMSs are more likely to experience negative cognitions, emotional distress and engage in maladaptive behavioural responses when their schemas are activated by stressful or schema congruent contexts in their environment [7]. For example, people who endorse schemas related to rejection are predicted to anticipate rejection, experience greater emotional distress and engage in maladaptive behavioural responses when presented with situations involving rejection (i.e., schema congruent

contexts). However, it is also likely that greater interpersonal problems give rise to more mal-adaptive beliefs and expectations about others, meaning that interpersonal problems could increase EMSs endorsement [7]. Given, EMSs and interpersonal problems might impact one another, further experimental research is needed to investigate Young, Klosko and Weishaar's [7] theory that EMSs increase problematic interpersonal responses.

One of the challenges of assessing a causal relationship between EMSs and problematic interpersonal responses is that individual differences such as EMSs are challenging to manipu-late in an experiment. However, given EMSs are theorised to increase problematic interper-sonal responses in stressful or schema congruent situations [7], we would expect to find people with high levels of EMSs to show increased problematic interpersonal responses in situations where there is an interpersonal stressor or high degree of congruence between the situation and the schema. In contrast, according to the Schema Therapy model [7] we would expect to find people with high EMSs to show no differences in interpersonal responses when compared to people with low EMSs in situations with low levels of interpersonal stress or low schema congruence. Therefore, while we are unable to manipulate EMSs, researchers could manipu-late the interpersonal situation and assess if EMSs moderate the relationship between the inter-personal situation and any problematic interpersonal responses.

Manipulating the interpersonal situation in the investigation of schemas was used in early research predating Young, Klosko and Weishaar's [7] theory of EMSs. For example, Koch used lexical decision tasks to assess if people reporting higher expectations about rejection were faster at identifying words related to rejection when compared to words related to accep-tance [23]. In this study, Koch [23] found that people with high levels of rejection schema were faster to identify rejection words after being primed with rejection words while people with low levels of rejection schema showed no differences in speed of identifying rejection words regardless of priming. In another study, Baldwin and Meunier [24] found that people with anxious attachment orientations responded faster to rejection words when primed with reject-ing interpersonal scenarios when compared to people with more secure attachment orienta-tions. These findings suggest that manipulating the interpersonal situation allows researchers an opportunity to assess how responses to interpersonal situations can be altered in people with higher or lower levels of EMSs. However, one disadvantage of lexical decision tasks is that responses are measured as reaction times to identifying words which provide little insight into the complex range of cognitive, emotional and behavioural responses a person might have in response to a challenging interpersonal situation.

## Interpersonal responses and impact of context

Different kinds of problematic interpersonal responses to interpersonal situations have been studied in a number of cross-sectional studies. Pierce and Lydon [25] found that participants with negative expectations about interpersonal relationships reported increased emotional dis-tress and more avoidant coping when presented with a written interpersonal scenario about a stressful interpersonal situation. Participants with negative interpersonal expectations also reported increased emotional distress, negative attributions and maladaptive coping responses after reading through interpersonal vignettes that depicted negative behaviour from a roman-tic partner [26]. Together, these studies suggest that people with negative beliefs or expecta-tions tend to show increased negative cognitions, distress and maladaptive behaviour in response to stressful interpersonal situations. However, these studies did not attempt to con-trol for, or manipulate, the interpersonal context [25, 26]. This limitation prevents researchers from drawing any causal conclusions between the interpersonal context and interpersonal responses. To address these limitations, the above mentioned schema activation paradigms

such as those utilised in early lexical decision tasks [23, 24] and measures of interpersonal responses such as those assessed by interpersonal vignettes [25, 26] could be combined by manipulating the interpersonal context using a series of written vignettes to assess if EMSs moderate the impact of interpersonal situations on interpersonal responses. Such an experimental design could assess if EMSs are associated with changes in the way we respond to challenging interpersonal situations.

Understanding how people with EMSs respond across different interpersonal contexts is also an area largely neglected by the research investigating EMSs and interpersonal problems [22]. The importance given to interpersonal contexts is argued by interpersonal theorists who have found that interpersonal behaviour is open to change across different contexts such as increased interpersonal pressures or responses from others [27]. For example, Farc et al. [28] found that people who experienced physical abuse during childhood were more likely to perceive aggression in children when the cues in the environment were ambiguous [28]. Similar findings have been found in other studies showing increased endorsement of hostility schemas to be associated with increased perceptions of hostile intentions in others when information about another person's behaviour was ambiguous [29]. Furthermore, people with higher rejection expectations reported increased withdrawal from a group when rejection in an interpersonal scenario was unambiguous compared to ambiguous [30, 31]. Together these findings suggest that individuals with EMSs are more likely to report negative responses (e.g. negative cognitions and behavioural responses) to interpersonal contexts depicting rejection and that responses are likely to become increasingly negative as the ambiguity about rejection in the scenario decreases. These findings are consistent with theoretical models depicting schemas (e.g. Schema therapy model, the rejection sensitivity model) arguing that people with stronger negative beliefs or schemas of themselves and others are more likely to perceive rejection in the environment and that perceptions of rejection are associated with higher levels of distress and more negative interpersonal behaviour responses [7, 31]. Relationship contexts such as whether a relationship is romantic, friendly or familial might also impact problematic interpersonal responses in people with EMSs [22]. For example, Fuhrman, Flannagan and Matamoros [32] argued that people have significantly higher expectations about their romantic partner's behaviour when compared to friends and family. However, most studies investigating the relationship between EMSs and interpersonal problems have been restricted to self-report questionnaires that measure general trait interpersonal tendencies devoid of any context [22]. Therefore, controlling for or comparing interpersonal responses across different contexts can provide researchers with more information about the role of EMSs in interpersonal problems.

## Study aims and hypotheses

The aim of the present study was to develop a new experimental paradigm to assess if EMSs moderate the impact of interpersonal contexts on interpersonal responses. Given that individuals with rejection schema have been found to have stronger associations with interpersonal problems [22], degree of rejection was manipulated in a series of interpersonal vignettes depicting acceptance, ambiguous rejection and rejection. Measuring degree of rejection based on acceptance, ambiguous and unambiguous rejection conditions is consistent with previous studies investigating cognitive, emotional and behavioural responses in people with negative relationship expectancies such as rejection sensitivity and hostility schema [e.g. 33, 35]. The present study's design also controlled for responses to scenarios depicting friends, family members and romantic partners. Based on the Schema Therapy model proposed by Young, Klosko and Weishaar [7] and earlier research investigating relational schemas, the following hypotheses were formulated:

Hypothesis 1: Participants will report increased negative interpersonal responses including negative cognitions, emotional distress and behavioural responses as degree of rejection increases across interpersonal scenarios.

Hypothesis 2: EMSs will moderate the relationship between the degree of rejection and negative interpersonal responses including negative cognitions, emotional distress and behavioural responses. That is, participants reporting higher EMSs will report higher levels of negative responses and participants reporting lower EMSs will report lower levels of negative responses to interpersonal rejection as the degree of rejection increases.

## Method

### Sample size

This research was conducted with approval from the University of New England Human Research Ethics Committee (Approval Number: HE18-248) with all participants providing informed and signed written consent for participation. We conducted a multi-level power analysis using the computer software MLPowSim [33] for a linear mixed model with 13 predictors including differences across individual vignettes, rejection conditions, total EMSs score and the interaction between EMSs and rejection conditions. With a type I error rate of 5% and power greater than 90% we estimated 150 participants were required for the present study.

### Participants

Participants were recruited from the first-year psychology student pool at the University of New England, Australia. The University of New England is a regional university with many mature age students who attend externally across Australia. Any person who was living in Australia, spoke English and was 18-years or older was allowed to participate in the present study. Participants were given a $15 supermarket (i.e., Coles) gift card for participation. In total, 176 participants were recruited for the experiment. Eighteen participants did not complete the entire experiment so they were excluded from the dataset due to missing data. As a result, the study sample consisted of 158 participants ($n = 43$, 27.2% male; $n = 114$, 72.2% female; $n = 1$, 0.6% other). Participant ages ranged from 19 to 72 ($M = 39.53$, $SD = 10.44$, eight participants did not report their age). Twenty-nine participants were single (18.4%), 40 participants were in a de-facto relationship (25.3%), 66 participants were married (41.8%), five participants were separated (3.2%), 14 participants were divorced (8.9%), and four participants were widowed (2.5%). Four participants reported that their highest level of education achieved was 11 years of schooling (2.5%), 25 had completed 13 years of schooling (15.8%), 28 participants completed a vocational qualification (17.7%), 78 had completed a bachelor's degree (49.4%), 21 had completed a master's degree (13.3%), and two had completed a doctorate (1.3%).

### Measures

**Young schema questionnaire-short form-third edition (YSQ-SF-3.** The YSQ-SF-3 [34] is a 90-item questionnaire that assesses the 18 EMSs identified by Young, Klosko and Weishaar [7]. Responses are rated on a 6-point Likert scale from 1 (*Entirely untrue of me*) to 6 (*Describes me perfectly*) where each EMS is scored based on the mean response across five questions. Mean scores for each EMS subscale range from one to six where scores between zero and three are considered of no clinical significance and scores greater than or equal to four are considered clinically significant [34, 35]. Scores across the 18 EMSs can also be used to calculate five domain scores by averaging scores for EMSs that share common themes (e.g., rejection and

disconnection). However, there is currently no guideline or normative data available for the interpretation of EMS domains or overall EMS scores. The YSQ-SF-3 has demonstrated good construct validity, internal consistency ($\alpha$ = .76 and $\alpha$ = .84 across EMS domains), and test-retest reliability ($r$ = .82) in adults [36, 37]. The present study found the YSQ-SF-3 to have high internal consistency ($\alpha$ = .97). The overall EMSs score which measures the total endorsement of all 18 EMSs was used in the present study's analysis which is often used in research [10, 38, 39] and was found to be highly correlated with rejection sensitivity in this study ($r$ = .48, $p$ < .001).

**Inventory of interpersonal problems (IIP-32).** The IIP-32 assesses trait problematic interpersonal behaviour patterns [40]. The IIP-32 consists of 32 items in the form of statements about the self that are rated on a 5-point Likert scale ranging from 1 (*not at all*) to 5 (*very much*). The IIP-32 measures dispositional interpersonal difficulty across two dimensions of affiliation (warmth/coldness) and agency (dominance/submissiveness) as suggested by the Interpersonal Circle or Circumplex model of interpersonal behaviour [41]. The IIP-32 splits these dimensions into eight octants or combinations of affiliation and agency including domineering, intrusive, nurturing, exploitable, non-assertive, socially avoidant, cold, and vindictive interpersonal behaviour, which can also be combined to produce an overall score of interpersonal problems. The IIP-32 also provides T-scores based on a normative sample of 800 adults aged 18 to 89-years-old for clinical interpretation of scores across subscales [42]. The IIP-32 has a good factor structure, test-retest reliability ($r$ = .90), an excellent overall internal consistency ($\alpha$ = .95) and internal consistency across subscales ranging from $\alpha$ = .82 to $\alpha$ = .94 [40]. The present study found the IIP-32 to have good internal consistency overall ($\alpha$ = .89).

**Rejection sensitivity questionnaire (RSQ).** The RSQ measures rejection sensitivity by asking participants to respond to a series of nine brief interpersonal scenarios [43]. For each scenario participants are asked to rate how concerned they are about their request being rejected (rejection concern) and how much they would expect the person in the scenario to accept their request (rejection acceptance). Participants rate their responses on a 6-point Likert scale for rejection concern from 1 (*very unconcerned*) to 6 (*very concerned*) while responses for rejection acceptance range from 1 (*very unlikely*) to 6 (*very likely*). Rejection acceptance scores are converted into rejection expectancy scores by subtracting rejection acceptance scores from seven [43]. Scores of rejection concern and rejection expectancy are then multiplied to produce an overall score of rejection sensitivity [43]. This overall rejection sensitivity score can be interpreted by comparing the deviation of the score from a mean rejection sensitivity score based on a sample of 685 healthy adults who completed the questionnaire electronically ($M$ = 8.61, $SD$ = 3.61; [43]). The RSQ has high test-retest reliability ($r$ = .91) and internal consistency ($\alpha$ = .89). The RSQ is also associated with insecure attachment, low self-esteem and neuroticism in addition to showing discriminant validity when comparing healthy adults to adults diagnosed with borderline personality disorder [44]. The present study found the RSQ to have good internal consistency ($\alpha$ = .84).

**Depression, anxiety and stress scale (DASS-21).** Psychiatric symptom distress was measured using the DASS-21 [45]. The DASS-21 is a widely used 21-item self-report scale measuring psychiatric symptom distress across three subscales of depression, anxiety and stress [45]. Symptoms are rated by frequency of occurrence over the past week on a 4-point Likert scale ranging from 0 (*never*) to 3 (*almost always*). Individual scores within each subscale are added to produce a total score which can fall between the "*normal*" to "*extremely severe*" range based on the deviation of scores from a normative sample of 1794 non-clinical adults [46]. The DASS-21 is a well-recognised scale for measuring psychiatric symptoms distress with good psychometric properties [45]. The present study found the DASS-21 to have high internal consistency ($\alpha$ = .94).

## Experiment stimuli

A series of nine interpersonal vignettes were written for the online experiment that depicted scenarios involving interpersonal rejection. To control for the type of relationship, three scenarios related to an interaction with a friend, three with a close family member, and three with a romantic partner. Each individual vignette had three variations of the scenario intended to provoke acceptance (control condition), ambiguous rejection (ambiguous condition), or clear rejection (rejection condition). Degree of rejection was manipulated in each of the nine interpersonal scenarios by changing the way the person in the scenario responded towards the participant including engaging with the participant's goals in the scenario (control), not engaging the participant's goals without providing information to identify a cause for non-engagement (ambiguous) and not engaging the participant with information suggesting that the person did not want to engage the participant (rejection). Each vignette and its variation across conditions is listed in S1 Appendix.

Responses to the interpersonal vignettes were broken down into questions about the cognitive, emotional and behavioural responses to each vignette. Participants were first asked to rate the degree of perceived rejection experienced on a sliding scale between 0 (*Not at all*) and 100 (*Severely rejected*) after reading each vignette both as a manipulation check and as a cognitive measure of perceived rejection. Negative cognitive responses were developed with reference to Weiner's [47] attribution theory which identifies three casual dimensions in attribution making: locus of control (internally caused or externally caused), stability and globality over an event. In the current experiment, participants were asked to rate the likelihood of any negative internal, external, intentional, global and stable attributions about the vignettes on a sliding scale from 0 (*Extremely unlikely*) to 100 (*Extremely likely*). In particular, participants were asked to rate the degree to which they perceived the scenario to be due to something negative about themselves (internal), something negative about the other person (external), the degree to which the person's behaviour was intentional or accidental (intentionality), unlikely to change (stability), and the degree to which they believed the behaviour negatively impacts other parts of their relationship (globality).

Emotional distress associated with each scenario was measured by using a subjective units of distress scale asking participants to rate the degree to which they would be angry or upset following the scenario on a sliding scale from 0 (*Not upset at all*) to 100 (*Extremely upset or angry*). However, unlike attributions and subjective measures of distress, behavioural responses in grounded theory are often defined by how disproportionate they are to the individual context (e.g. the schema coping style of overcompensation [7]) which meant questions related to specific behaviours would have needed to vary across scenarios making it difficult to differentiate whether changes in condition or changes in behavioural response questions predicted changes in behavioural responses. Due to this variability of behavioural responses across contexts, behavioural responses were measured qualitatively embedded within a predominantly quantitative design to keep questions across scenarios consistent. Behavioural responses were measured by asking participants to record a written response summarising how they would likely respond to the interpersonal situation and why. The behavioural responses were then coded using inductive content analysis to create a series of binary variables scoring endorsement or non-endorsement of each behavioural response so that behaviours could be assessed by statistical analysis with respect to the study hypotheses. To ensure participants followed experiment instructions, participants were asked if they answered vignettes based on how they think they would react rather than what they thought was the best response (Yes/No) at the conclusion of the experiment. Participants were also asked to rate how well they were able to visualise or put themselves in the scenarios on a sliding scale between 0 (*Extremely*

*difficult*) and 100 (*Extremely easy*). An example vignette and response layout is presented in S2 Appendix.

## Procedure

Participants were first sent a paper-based survey in the mail asking them to respond to some demographic questions and to complete the YSQ-SF-3 and IIP-32 due to the YSQ-SF-3 electronic distribution restrictions. Immediately after completing the paper-based survey, participants were directed to a Qualtrics [48] link to complete the online experiment and RSQ. Data from the paper-based questionnaires and online experiment results were linked using a unique identifier number.

The online experiment was a repeated measures design such that participants completed nine vignettes in total: three vignettes from each of the control, ambiguous and rejection conditions. Participants completed one friend, family and romantic vignette in every condition (i.e., control, ambiguous and rejection conditions). Participants only completed one variation of each vignette to prevent participants from completing the same scenario more than once and keep participants blind to the manipulation of rejection conditions. Presentation of each condition (control, ambiguous and rejection) and type of relationship (friend, family and romantic partner) were randomised in presentation to control for any variation in responses due to order of condition or presentation of scenarios depicting a specific type of relationship. The variation of vignette used in each condition was also randomised to control for any variability in responding across different scenarios. An example of the presentation of conditions and vignettes and allocation to respective presentation order groups is depicted in Fig 1.

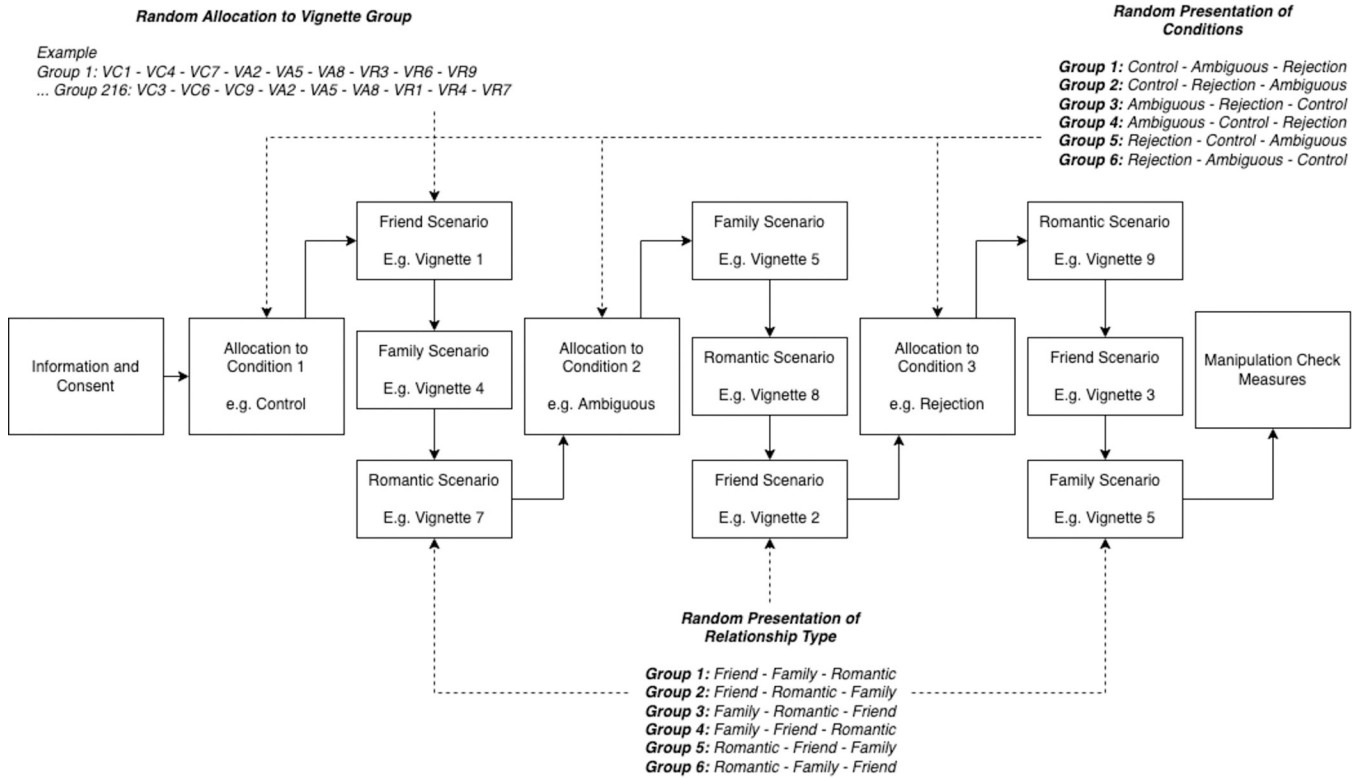

**Fig 1. Example summary of experiment depicting the presentation of conditions and vignettes.**

## Coding and scoring

**Quantitative coding and scoring.** As the observations were hierarchical in nature (multiple observations from a participant nested within that participant), we adopted a hierarchical approach. The dataset comprised of two sources of variance: the within-participant responses to each of the nine scenarios for each participant (Level 1) and the between-participant EMS scores (Level 2). The within-participant variance refers to how the responses between each of the scenarios varied within each participant.

Rejection condition was dummy coded as a categorical variable within each participant (Level 1) while EMS score on the YSQ-SF-3 was coded as a continuous variable that only varied between participants (Level 2). The EMS scores (YSQ-SF-3) were centred based on the grand mean (Level 2) prior to interpreting the results of the statistical analysis consistent with the recommendations of Aguinis, Gottfredson and Culpepper [49] when testing cross-level moderations. Scores for each individual scenario were averaged across the control, ambiguous and rejection scenarios to produce a vignette group control variable which accounted for the variance due to differences in vignette scenarios and relationships depicted in the vignettes (i.e., friend, family and romantic partner scenarios).

**Qualitative coding of behavioural responses.** Qualitative behavioural responses were coded using inductive content analysis as described by Elo and Kyngas [50]. The two qualitative responses *"What would you do"* and *"Why would you respond this way"* were combined and treated as a single response as recommended by Braun et al. [51] for online open-ended qualitative questions. Responses for each vignette were coded separately. A single clause was considered the unit of analysis meaning clauses divided by conjunctions were coded separately. Multiple codes were given for responses where different clauses suggested different responses. Responses were first read through by the lead researcher (coder) and an independent researcher (reviewer) to familiarise the researchers with the data prior to beginning coding. During this process, it was observed that while the question "*What would you do?"* aimed to assess behavioural responses, participants also answered questions by referring to cognitions and emotions. For these reasons, codes were separated into cognitions, emotions and actions. Where there was ambiguity about whether a response was a cognition, emotion or action (e.g., disappointment could be considered a cognition or emotion) the response was coded under all relevant categories.

The inductive content analysis started with open coding by the coder where all responses were read through individually and codes were created that represented the actions described. All responses for each code were collated so that all responses within each code could be reviewed collectively. At the completion of the open coding stage there were 229 codes for cognitions, 48 codes for emotions and 100 codes for actions.

After the open coding stage, responses were reviewed to cross check if they collectively fit within each code. New codes were created for responses that did not match the other responses and codes were combined where there was conceptual overlap. During the second stage of coding, all similar codes were grouped into second-order categories by grouping any similar codes together (e.g., the situation avoidance category was formed by combining the codes avoid situation, cancel plans with other person and end conversation). Further abstraction of second-order categories was achieved by grouping any related second-order categories into third-order categories. This was completed by grouping any categories that were conceptually similar (e.g., avoiding the interaction, avoiding the situation and avoiding the person) into third-order categories (e.g., avoidance). At the end of the abstraction stage the reviewer reviewed all the responses and first to third-order codes and categories. Codes and categories were discussed between the coder and reviewer and refined to ensure conceptual

distinctiveness and that they accurately represented the data, before agreeing on the final set of codes and categories. For example, categories that could fit into multiple categories due to vagueness in the response (e.g., "I would confront them") were discussed between the coder and reviewer until a consensus was reached about the most appropriate category for that response (e.g., fighting). Given we were exclusively reviewing the qualitative responses to identify behavioural responses to interpersonal scenarios, only the behavioural responses were reported in this study.

## Statistical analysis

Given the hierarchical structure of the data, multi-level mixed effect linear and logistic regression models were used to investigate the moderating effect of EMSs (W) on the relationship between degree of rejection (X) and negative responses to interpersonal situations (Y). Multi-level modelling is recommended when assessing a moderator (EMSs) in a repeated measures design with a three-level categorical independent variable (degree of rejection) on a continuous or binary dependent variable (cognitive, emotional and behavioural responses to interpersonal situations; [52]). To assess the moderation of EMSs and rejection condition on each dependent variable we followed the recommendations of Aguinis, Gottfredson and Culpepper [49] but omitted the addition of a random slope due to the small number of observations within participants ($n = 9$) and non-convergence issues. While the omission of a random slope can inflate the type I error rate, its omission is considered appropriate where there are convergence issues [53]. Given the potential inflation of the type I error rate due to the omission of a random slope and multiple outcome variables the conservative Bonferroni procedure was used to adjust for the type I error rate as recommended by Frane [54]. Consequently, we adjusted our overall alpha level to $p = .003$ by dividing .05 by all 15 outcome variables and $p = .001$ for pairwise contrasts across the control, ambiguous and rejection conditions.

The analyses were conducted in Jamovi [55] using the GAMLj module for General Analyses of Linear Models [56] which is a robust module for estimating general linear, mixed and generalised multi-level models [57, 58]. A two-level random effects model using the restricted maximum likelihood (REML) method was estimated for each continuous dependent variable including perceived rejection, negative cognitions, emotional distress and behavioural responses. Independent variables of vignette group, rejection condition, EMS score and the cross-level interaction of EMSs and rejection were added to the model in a stepwise process as recommended by Aguinis, Gottfredson and Culpepper [49].

In step one, the null model with a random intercept was estimated for each dependent variable to determine if multi-level modelling was appropriate [49]. The ICC values for perceived rejection, cognitions and emotional distress were all greater than .1 suggesting it was appropriate to proceed with multi-level modelling. However, all null model ICC for qualitatively coded actions (excluding fighting) were lower than .1 suggesting limited dependency within participants for the qualitatively coded actions. Given our level 1 variance comprised of repeated measures we continued with multi-level modelling as our design implied dependency of scores within participants and failing to control for the dependency of measures even where there are low ICC's can inflate the type I error rate [59].

In step two, vignette group was added to control for differences in the dependent variables due to differences in the interpersonal scenarios provided in each interpersonal vignette [49]. Vignette group variance included differences in vignettes due to scenarios depicting friends, family and romantic partners. We were unable to control for vignette group in the mixed binary logistic regressions for the qualitatively coded actions due to non-convergence issues associated with having too many categorical variables predicting a single binary outcome variable.

In step three, rejection condition (i.e., control, ambiguous and rejection) was dummy-coded and added to assess if degree of rejection significantly predicted changes in the dependent variables. Simple contrast $t$-tests were conducted to assess the differences between all three conditions using Bonferroni adjusted p-values.

In step four, total EMS score on the YSQ-SF-3 was added to the model to assess if EMS scores predicted changes in each of the dependent variables [49]. In the final step of model building, the interaction between rejection condition and EMS score was added to assess the cross-level moderation effect of (high and low) levels of EMSs on the relationship between rejection condition and interpersonal responses [49]. For the logistic regression analyses the estimated probability of endorsing each action in each condition was reported using the $P$ symbol. Log-likelihood ratio tests were performed to assess if the addition of each predictor at each step of the model building process significantly explained a unique proportion of model variance. The log-likelihood ratio tests were one-tailed due to the variance only being able to deviate from zero in a positive direction [60].

We computed Cohen's $f^2$ for both the marginal and conditional changes in $R^2$ as a measure of effect size for each fixed effect as it was added to the model including vignette group, rejection condition, EMS score and the interaction between rejection condition and EMS scores. The effect size for the change in marginal $R^2$ was denoted by $f^2_m$ which describes the effect size for the change in the between participant variance (Level 2) explained by the fixed effect [61]. The effect size for the change in conditional $R^2$ was denoted by $f^2_c$ which describes the effect size for the change in the combined between and within participant variance (Level 1 and Level 2) explained by the fixed effect [61]. The results for the two-level random effects models and effect sizes for each dependent variable of perceived rejection, negative cognitions, emotional distress and behavioural responses are described below and summarised in Tables 1 and 2.

## Results

### Manipulation and assumption checks

Manipulation check measures were reviewed with 96.8% ($N = 153$) of participants reporting they responded to vignettes based on how they think they would act rather than what they felt was the best course of action. Participants reported the mean ability to visualize themselves in the vignettes to be $M = 80.3$ ($SD = 20.3$) on a scale between 0 (*Extremely difficult*) and 100 (*Extremely easy*). Significant differences in perceived rejection were found between the different vignette scenarios ($F(8, 1252) = 14.56$, $p < .001$, $f^2_m < .01$, $f^2_c = .01$) suggesting that it was appropriate to control for differences in vignettes. Participants reported increased levels of perceived rejection as the degree of rejection in the interpersonal scenarios increased ($F(2, 1252) = 436.75$, $p < .001$, $f^2_m = .12$, $f^2_c = .31$). Participants reported higher levels of perceived rejection in the ambiguous scenarios ($M = 31.59$, $SE = 1.36$, $t(1252) = -18.77$, $p < .001$) and rejection scenarios ($M = 46.36$, $SE = 1.37$, $t(1252) = -29.16$, $p < .001$) when compared to the control scenarios ($M = 4.92$, $SE = 1.37$). Participants also reported higher levels of perceived rejection in the rejection scenarios when compared to the ambiguous scenarios ($t(1252) = -10.40$, $p < .001$).

Perceived rejection, self-blame, other-blame, intentionality, stability, globality and emotional distress for each vignette were added together to produce a total score for each participant across the entire experiment. Table 3 shows that perceived rejection in scenarios and interpersonal response variable scores were correlated with similar measures including rejection sensitivity scores (RSQ), trait interpersonal problems (IIP) measures of depression, anxiety and stress (DASS-21).

**Table 1. Regression output for the effect of vignette group, rejection condition and EMSs on cognitive and emotional responses to interpersonal scenarios.**

| Level and Variable | Percieved Rejection | Self-Blame | Other-Blame | Perceived Intentionality | Perceived Stability over Time | Percieved Globaility | Emotional Distress |
|---|---|---|---|---|---|---|---|
| **Level 1** | | | | | | | |
| **Intercept ($\gamma_{00}$)** | 1.43 (2.12) | 10.00 (1.95)*** | 11.12 (2.30)*** | 51.55 (2.93)*** | 44.54 (2.75)*** | 9.20 (2.35)*** | 0.64 (2.04) |
| **Vignette Group**<br>**Vignette 1 –Friend**<br>**Vignette 2 –Friend ($\gamma_{1,0}$)**<br>**Vignette 3 –Friend ($\gamma_{2,0}$)**<br>**Vignette 4 –Family ($\gamma_{3,0}$)**<br>**Vignette 5 –Family ($\gamma_{4,0}$)**<br>**Vignette 6 –Family ($\gamma_{5,0}$)**<br>**Vignette 7 –Romantic Partner ($\gamma_{6,0}$)**<br>**Vignette 8 –Romantic Partner ($\gamma_{7,0}$)**<br>**Vignette 9 –Romantic Partner ($\gamma_{8,0}$)** | $F(8, 1252) =$ 14.56*** Reference Group 5.69 (2.46)* 7.51 (2.46)** -9.30 (2.46)*** 10.00 (2.46)*** 2.37 (2.46) -1.05(2.46) 13.06 (2.46)*** 3.10 (2.46) | $F(8, 1252) =$ 8.09*** Reference Group 4.11 (2.27) 1.77 (2.27) -8.44 (2.27)*** -1.80 (2.27) -6.75 (2.27)** 4.64 (2.27)* -0.66 (2.27) -4.39 (2.27) | $F(8, 1252) =$ 6.69*** Reference Group 4.04 (2.67) -1.76 (2.67) -10.09 (2.67)*** 3.43 (2.67) -4.56 (2.67) -5.38 (2.67)* 4.05 (2.67) -3.67 (2.67) | $F(8, 1252) =$ 17.22*** Reference Group 13.79 (3.42)*** 11.33 (3.42)*** -7.86 (3.42)* 5.42 (3.42) -9.08 (3.42)** 17.80 (3.42)*** -7.94 (3.42)* 3.19 (3.42) | $F(8, 1252) = 4.46***$ 14.83 (3.29)*** 11.77 (3.29)*** 5.51 (3.29) 10.88 (3.29)*** 2.49 (3.29) 4.42 (3.29) 3.25 (3.29) 9.51 (3.29)** | $F(8, 1252) =$ 5.85*** Reference Group 8.64 (2.74)** 2.45 (2.74) -4.50 (2.74) 5.88 (2.74)* -2.24 (2.74) 0.38 (2.74) 9.08 (2.73)*** 1.61 (2.74) | $F(8, 1252) =$ 14.77*** Reference Group 4.19 (2.33) 6.69 (2.33)** -7.45 (2.33)** 7.96 (2.33)*** 7.90 (2.33)*** 0.84 (2.33) 14.66 (2.33)*** 0.63 (2.33) |
| **Degree of Rejection**<br>**No Rejection**<br>**Ambiguous Rejection ($\gamma_{9,0}$)**<br>**Clear Rejection ($\gamma_{10,0}$)** | $F(2, 1252) =$ 436.75*** Reference Group 26.67 (1.42)*** 41.44 (1.42)*** | $F(2, 1252) =$ 101.50*** Reference Group 12.65 (1.31)*** 18.22 (1.31)*** | $F(2, 1252) =$ 198.41*** Reference Group 20.41 (1.54)*** 30.10 (1.54)*** | $F(2, 1252) =$ 62.65*** Reference Group -21.65 (1.97)*** -7.12 (1.97)*** | $F(2, 1252) =$ 60.98*** Reference Group -19.68 (1.90)*** -16.14 (1.90)*** | $F(2, 1252) =$ 162.90*** Reference Group 18.61 (1.58)*** 28.01 (1.58)*** | $F(2, 1252) =$ 466.88*** Reference Group 28.39 (1.34)*** 39.90 (1.34)*** |
| **Level 2** | | | | | | | |
| **EMS Endorsement ($\gamma_{0,1}$)** | $F(1, 156) =$ 34.33*** 0.03 (0.02) | $F(1, 156) =$ 53.25*** 0.06 (0.02)** | $F(1, 156) =$ 13.16*** 0.04 (0.02) | $F(1, 156) = 9.24**$ 0.07 (0.03)* | $F(1, 156) = 25.22***$ 0.10 (0.03)*** | $F(1, 156) =$ 19.55*** 0.05 (0.03) | $F(1, 156) =$ 24.85*** 0.03 (0.02) |
| **Cross-level interaction** | | | | | | | |
| **Rejection*EMS Endorsement**<br>**No Rejection*EMS**<br>**Ambiguous Rejection*EMS ($\gamma_{9,1}$)**<br>**Clear Rejection*EMS ($\gamma_{10,1}$)** | $F(2, 1252) =$ 18.43*** Reference Group 0.11 (0.02)*** 0.14 (0.02)*** | $F(2, 1252) =$ 14.00*** Reference Group 0.06 (0.02)** 0.11 (0.02)*** | $F(2, 1252) =$ 2.11 Reference Group 0.03 (0.03) 0.05 (0.03)* | $F(2, 1252) = 0.91$ Reference Group -0.002 (0.03) 0.01 (0.03) | $F(2, 1252). = 1.91$ Reference Group -0.02 (0.03) 0.04 (0.03) | $F(2, 1252) =$ 4.67* Reference Group 0.04 (0.03) 0.08 (0.03)** | $F(2, 1252) =$ 12.64*** Reference Group 0.10 (0.02)*** 0.09 (0.02)*** |
| **Variance components** | | | | | | | |
| **Within participant (L1) variance ($\sigma^2$)** | 478.05 | 407.56 | 563.27 | 920.86 | 854.40 | 590.56 | 427.58 |
| **Between participant (L2) variance ($\tau_{00}$)** | 135.76 | 107.53 | 153.74 | 244.95 | 162.12 | 160.04 | 138.40 |
| **ICC** | 0.22 | 0.21 | 0.21 | 0.21 | 0.16 | 0.21 | 0.25 |
| **-2 log likelihood (FIML)** | 12979.68*** | 12743.50*** | 13206.34 | | | | 12838.35** |
| **Number of estimated parameters** | 16 | 16 | 16 | 16 | 16 | 16 | 16 |
| **Conditional $R^2$ (Total Model Fit)** | 0.52 | 0.37 | 0.38 | 0.32 | 0.26 | 0.37 | 0.54 |

Note. EMS = Early Maladaptive Schema score; ICC = Intraclass correlation coefficient; FIML = Full information maximum likelihood estimation; L1 = Level 1;

L2 = Level 2; $N_{L1} = 1467$; $N_{L2} = 163$. Values in parentheses report standard errors.

*$p < .05$, **$p < .01$, ***$p < .001$

The assumptions of normality and linearity were satisfied in addition to model variables not showing any signs of multicollinearity. Visual inspection of the scatterplot of residuals suggested signs of double outward box heteroscedasticity which was confirmed by significant log-likelihood ratio test differences when comparing the random-intercept only model of the absolute residuals to the full model of absolute residuals regressed on all of the predictors [62].

**Table 2. Regression output for the effect of rejection condition and EMSs on qualitatively coded behavioural responses to interpersonal scenarios.**

| Level and Variable | Engaging | Collaborating | Supporting | Avoiding | Fighting | Expressing positive affect | Expressing negative affect | Non-engagement |
|---|---|---|---|---|---|---|---|---|
| **Level 1** | | | | | | | | |
| Intercept ($\gamma_{00}$) | 1.74 (0.24)*** | -1.85 (0.26)*** | -2.10 (0.19)*** | -3.07 (0.39) | -5.94 (1.27) | -0.29 (0.17) | -4.67 (0.90) | -0.82 (0.18) |
| Degree of Rejection<br>No Rejection<br>Ambiguous Rejection ($\gamma_{9,0}$)<br>Clear Rejection ($\gamma_{10,0}$) | $\chi^2(2)$ = 68.86***<br>Ref<br>-1.20 (0.28)***<br>-2.57 (0.31)*** | $\chi^2(2)$ = 66.62***<br>Ref<br>2.74 (0.34)***<br>2.22 (0.32)*** | $\chi^2(2)$ = 41.00***<br>Ref<br>-0.15 (0.37)<br>1.62 (0.32)*** | $\chi^2(2)$ = 50.87***<br>Ref<br>2.53 (0.41)*** | $\chi^2(2)$ = 21.12***<br>Ref<br>3.01 (1.21)*<br>4.34 (1.21)*** | $\chi^2(2)$ = 57.89***<br>Ref<br>-3.04 (0.51)***<br>-2.46 (0.41)*** | $\chi^2(2)$ = 15.79***<br>Ref<br>3.15 (0.92)***<br>2.25 (0.95)* | $\chi^2(2)$ = 3.35<br>Ref<br>-0.07 (0.25)<br>0.34 (0.24) |
| | | | | 3.02 (0.42)*** | | | | |
| **Level 2** | | | | | | | | |
| EMS Endorsement ($\gamma_{0,1}$) | $\chi^2(1)$ = 0.35<br>-0.002 (0.004) | $\chi^2(1)$ = 1.23<br>-0.005 (0.004) | $\chi^2(1)$ = 0.04<br>-0.001 (0.004) | $\chi^2(1)$ = 0.27<br>-0.003 (0.006) | $\chi^2(1)$ = 0.36<br>0.009 (0.02) | $\chi^2(1)$ = 0.70<br>0.002 (0.003) | $\chi^2(1)$ = 0.14<br>0.01 (0.01) | $\chi^2(1)$ = 0.04<br>0.001 (0.003) |
| **Cross-level interaction** | | | | | | | | |
| Rejection*EMS Endorsement<br>No Rejection*EMS<br>Ambiguous Rejection*EMS ($\gamma_{9,1}$)<br>Clear Rejection*EMS ($\gamma_{10,1}$) | $\chi^2(2)$ = 0.12<br>Ref<br>0.001 (0.004)<br>0.001 (0.005) | $\chi^2(2)$ = 3.82<br>Ref<br>0.0003 (0.005)<br>-0.008 (0.005) | $\chi^2(2)$ = 1.22<br>Ref<br>-0.003 (0.006)<br>0.003 (0.005) | $\chi^2(2)$ = 3.14<br>Ref<br>0.01 (0.007)<br>0.012 (0.007) | $\chi^2(1)$ = 0.34<br>Ref<br>-0.007 (002)<br>-0.004 (002) | $\chi^2(2)$ = 4.88<br>Ref<br>0.008 (0.007)<br>-0.01 (0.007) | $\chi^2(2)$ = 0.71<br>Ref<br>-0.008 (0.01)<br>-0.005 (0.01) | $\chi^2(2)$ = 1.60<br>Ref<br>0.002 (0.004)<br>-0.003 (0.004) |
| **Variance components** | | | | | | | | |
| Within participant (L1) variance ($\sigma^2$) | 1.00 | 1.00 | 1.00 | 1.00 | 1.00 | 1.00 | 1.00 | 1.00 |
| Between participant (L2) variance ($\tau_{00}$) | 0.18 | 0.61 | 0.12 | 0.60 | 1.59 | 0.14 | <0.01 | 0.12 |
| ICC | 0.05 | 0.16 | 0.04 | 0.20 | 0.33 | 0.04 | <0.01 | 0.04 |
| -2 log likelihood (FIML) | 541.38*** | 530.38*** | 425.05*** | 482.07*** | 272.33*** | 347.18*** | 260.84*** | 596.74 |
| Number of estimated parameters | 8 | 8 | 8 | 8 | 6 | 8 | 8 | 8 |
| Conditional $R^2$ (Total Model Fit) | 0.28 | 0.41 | 0.19 | 0.43 | 0.61 | 0.39 | 0.38 | 0.05 |

Note. EMS = Early Maladaptive Schema score; ICC = Intraclass correlation coefficient; FIML = Full information maximum likelihood estimation; L1 = Level 1;

L2 = Level 2; NL1 = 489; NL2 = 163. Values in parentheses report standard errors.

*$p < .05$

**$p < .01$

***$p < .001$

Such heteroscedasticity is common in experiments with control conditions and likely reflected reduced variability of responses in the control scenarios when compared to the ambiguous and rejection scenarios [63]. We decided to proceed with interpretation of the multilevel models despite the observed heteroscedasticity to preserve simplicity and interpretability of data given previous simulation studies have found multi-level models to be robust to violations in homoscedasticity when there are equal observations per condition [62, 63].

## Qualitative coding of behavioural responses

Eight third order category actions were identified as responses to the interpersonal scenarios using inductive content analysis including (1) engaging, (2) collaborating, (3) supporting, (4) fighting, (5) avoiding, (6) affect expression, (7) uncertainty, and (8) non-engagement. Each of these categories is described below in further detail. Third-order behaviour categories for each

**Table 3. Correlations (Pearson _r_) between total interpersonal response scores and trait interpersonal problems, rejection sensitivity and psychiatric symptom distress.**

| Total Response to Interpersonal Scenarios | Interpersonal Problems (IIP) | Rejection Sensitivity (RSQ) | Symptoms of Depression (DASS-21) | Symptoms of Anxiety (DASS-21) | Symptoms of Stress (DASS-21) |
|---|---|---|---|---|---|
| **Perceived Rejection** | .42*** | .50*** | .31*** | .30*** | .49*** |
| **Self-blame** | .47*** | .52*** | .41*** | .44*** | .52*** |
| **Other-blame** | .26*** | .33*** | .13 | .20* | .30*** |
| **Intentionality** | .13 | .09 | .16* | .14 | .28*** |
| **Stability** | .25** | .25*** | .25** | .18* | .34*** |
| **Globality** | .32*** | .41*** | .19* | .23** | .34*** |
| **Emotional Distress** | .36*** | .43*** | .22*** | .25*** | .42*** |
| **Total EMSs Score** | .66*** | .48*** | .66*** | .55*** | .67*** |

Note. EMS = Early Maladaptive Schema

*$p < .05$

**$p < .01$

***$p < .001$

scenario were then coded into binary variables as 1 (the behavioural response reported in the scenario) or 0 (the behaviour response not reported in the scenario). The second and third-order categories with specific examples are listed in Table 4. The percentages of participants who reported engaging in each third-order category action across each condition (i.e., control, ambiguous and rejection) are reported in Table 5.

**Engaging.** The engaging category encompassed actions where participants showed an interest in pursuing social interaction or desire to spend time with the person depicted in the scenario. This included actions that were focused on initiating or engaging in casual interaction such as "talking" or "sharing" information rather than addressing or resolving the perceived issue (Table 4).

**Collaborating.** The collaborating category encompassed actions that were intended to resolve issues presented in scenarios by communicating perceived issues with the other person and attempting to compromise on solutions. This included asking questions, listening and validating and sharing opinions or perspectives about the issues presented in the interpersonal vignettes (Table 4).

**Supporting.** The supporting category encompassed actions that appeared to be focused on resolving the issues presented in the scenarios by attempting to meet the other person's needs or by sacrificing one's own interpersonal goals due to the actions of the other person. This included offering to help the other person and offering to accommodate the other persons wishes in the scenario without pursuing the participant's goals in the scenario (Table 4). For example, in rejection vignette four (S1 Appendix) where participants have prearranged a time to talk with a family member who tells the participant they cannot talk, an accommodating response would include instructing the family member to do "whatever it is that they need to do, and to give me a call back when they have the time."

**Fighting.** The fighting category encompassed actions where participants prioritised their own needs, feelings and perspectives above those of the other person in the scenario. Fighting responses included actions such as arguing, retaliating or making negative comments directed at the other person based on negative attributions of the scenario that extended beyond what was indicated in the vignette (Table 4). For example, in control vignette seven (S1 Appendix) where the participant is greeted by their romantic partner by being asked what they would like

**Table 4. Qualitative categories for actions in response to interpersonal scenarios with examples, number of participants that endorsed the action (N) and their mean Inventory of Interpersonal Problem t-scores (IIP T-Score) and standard deviations (SD).**

| Third-Order Category | Second-Order Category | Codes | Endorsed (N) | Examples |
|---|---|---|---|---|
| Engaging | *Initiate interaction* | • Initiate contact<br>• Invite other person to do something<br>• Make an effort to increase contact with other person<br>• Try again to engage/re-engage<br>• Greet other person | 79 | Call them on the phone<br>I would try to contact them to organise something<br>try to keep regular contact with this family member<br>I would give them another call and see if we could schedule something<br>Tap them on ther shoulder and say hi |
| | *Engage activity with other person* | • Join person in an activity<br>• Plan an activity together<br>• Engage an activity with the other person<br>• Agree to spend time with other person<br>• Give suggestions to engage | 54 | join them for a drink.<br>do something fun together<br>We would do something fun together.<br>Agree and meet up<br>Suggest some ideas and decide what we want to do together |
| | *Engage in conversation* | • Engage in general conversation<br>• Engage in a conversation by venting<br>• Engage in conversation by sharing information about self<br>• Enjoy experience/activity<br>• Move in close proximity of other person | 118 | we would catch up<br>I would vent openly to them about my problems, and answer questions as they arose.<br>I would continue talking and sharing with them.<br>I would enjoy the conversation<br>lean or sit on him and ask him if he had any ideas |
| | *Reciprocate engagement from other person* | • Reciprocate engagement from other person<br>• Reciprocate contact from other person | 47 | I would be affectionate back<br>smile back |
| Collaborating | *Ask and Clarify* | • Clarify situation<br>• Ask other person to do something<br>• Ask for what is wanted<br>• Seek advice<br>• Enquire about other person<br>• Question why undesired behaviour occurred<br>• Ask to be included or invited | 152 | I would probably call them to ask why they couldnt come.<br>I would ask them to get off their phone and just enjoy the moment<br>Ask them to call back.<br>I'd keep chatting to them and seeking their advice<br>ask how they have been<br>I would write back and ask why they could not make it and why they could let me know earlier<br>I would ask if i can join them |
| | *Listen and validate* | • Give attention and listen<br>• Validate the other person<br>• Acknowledge the other person's needs<br>• Communicate respect for other person | 31 | I would listen and be open minded to what my husband wants to say to me<br>I would leave a message on there phone saying "hi, oh I am sorry we missed each other, I was so looking forward to talking to you. I understand how time can get away from us all though, but i hope you are going ok?<br>Acknowledged family member as being busy.<br>Respect my friend's wishes. |
| | *Talk about Issue* | • Talk to the other person about issue | 9 | I would talk to them—clearly its important otherwise they wouldn't have been acting weird/look preoccupied. |
| | *Explain own feelings and perspective* | • Explain own situation/perspective<br>• Explain negative feelings<br>• Communicate feelings and needs<br>• Remind other person of responsibility to you<br>• Explain what you have done for other person<br>• Reassure other person about yourself<br>• Give an opinion | 68 | Follow up with them afterward to express my disappointment<br>Explain that I was hurt they hadn't turned up through it myself<br>I would try talk to him about it to explain my feelings.<br>I will ring back a bit later and remind them about their commitment.<br>explain that I listened to them and deserve the same in return<br>just inform them I am ok.<br>I think that I would ask why this was his response. Did he realise that it would be good for us to have some time together? |
| | *Tell other person what is wanted* | • Tell other person what to do | 5 | tell them to stop what they are doing and to listen to me |

*(Continued)*

**Table 4.** (Continued)

| Third-Order Category | Second-Order Category | Codes | Endorsed (N) | Examples |
|---|---|---|---|---|
| | *Compromise* | • Compromise | 6 | I would ask when we could spend some time together as a compromise |
| **Supporting** | *Supporting others needs* | • Offer support and help <br> • Introduce other person to others | 2 | Offer help <br> make sure they have people to talk to if they don't know the crowd etc. |
| | *Putting others needs above own goals* | • Accommodate other person's needs <br> • Acknowledge the other person is busy and suggest alternative | 55 | I would go and put the kettle on for them to help them get ready <br> tell them if they are busy i can speak to them another time |
| | *Accepting* | • Accept situation | 42 | I would choose to just accept it and get on with things. |
| | *Forgiving* | • Forgive other person | 3 | I forgive him for being easily side tracked |
| **Fighting** | *Confrontation* | • Confront other person about perceived issue <br> • Argue with the other person <br> • Accuse other person of problematic behaviour | 30 | I would confront my friend <br> I would call and argue about probably why they didnt think of me that day. <br> I would say how rude that is and tell them they need to consider how their words and actions can hurt people |
| | *Retaliate* | • Retaliate to perceived negative actions | 3 | I can only respond to by ignoring them as well |
| | *Make negative comments* | • Make passive aggressive comment <br> • Make a comment based on negative perception of other person or situation | 26 | say, so you dont know me enough to plan things for us, or does it have to be me all the time? <br> Ask what their problem is, remove them from your life |
| **Avoiding** | *Avoiding Interaction* | • Avoid responding or sharing <br> • Not share information or disclose <br> • Avoid raising an issue despite reporting negative affect <br> • Limit interaction with other person <br> • Try to cover up actions as accidental <br> • Hide feelings or perspective <br> • Pretend as if there is no problem <br> • Ignore the other person's behaviour <br> • Sulk <br> • Repeatedly go over scenario in head (e.g. ruminate, overthink) | 42 | I would make a note not to vent to that person again in future <br> I would remember this event and be a little less open and enthusiatic with this friend. <br> I would keep how I was feeling to myself, as it wouldnt change things with them anyway. <br> I'd leave them alone and hope they make it up to me later or at least have a good reason, if they don't I'd probably be a bit short with them. <br> I may try messaging again a long time later making it look like I wanted to talk about something else <br> I would say that it is ok to them, but would cry and be upset <br> Keep everything to myself, put on a happy face because they're either busy or not in the mood to listen to my problems. <br> I would ignore my friend at the time and make them aware I am not happy. <br> sulk and stew for a while. <br> I would dwell on this for a very, very long time. |
| | *Avoiding Situation* | • Avoid situation (e.g. interacting) <br> • Cancel plans with other person <br> • End conversation | 52 | I would cancel my planning for the day with him. <br> Hang up <br> Pretend did not see them. |
| | *Avoiding Person* | • Avoid person <br> • Stop contacting the other person <br> • End relationship <br> • Keep walking/walk away from other person <br> • Give space to the other person <br> • Raise issue with someone else | 88 | I would avoid them. <br> Stop the texting and calling <br> Probably break up with them. <br> walk away <br> Leave the room/house and do something on my own. <br> I might have a whinge to my partner or siblings about it |
| **Expressing positive affect** | *Express appreciation* | • Thank <br> • Express liking of situation <br> • Express appreciation for other person | 38 | I would thank my friend <br> I'd send a text message to say I had a great time <br> I would tell them I loved them and am grateful for them. |
| | *Express positive affect* | • Express positive affect <br> • Express physical affection | 17 | Show appreciation and excitement <br> I would hug them |
| | *Joke* | • Make a joke | 14 | Probably joke about how hard it can be to spot things sometimes that are right in front of your nose |

*(Continued)*

**Table 4.** (Continued)

| Third-Order Category | Second-Order Category | Codes | Endorsed (N) | Examples |
|---|---|---|---|---|
| **Expressing negative affect** | *Express negative affect* | • Over-react<br>• Express negative affect (e.g. cry, worry)<br>• Non-verbally express confusion | 20 | I would probably over react<br>I would panic, over think and cry<br>show being puzzled about seeing her out and about |
| | *Seek reassurance* | • Express concern for other person<br>• Seek reassurance about worry<br>• Repeatedly attempt to contact | 28 | Leave a message and let them know that I called, I hope your okay but we were going to talk. Call me back when you can<br>I would get upset at them as this is not acceptable as I would be concerned to see if they are ok<br>I would keep trying to contact them |
| **Uncertainty** | *Uncertainty about how to respond* | • Hesitate/unsure of response<br>• Not sure | 2 | hesitate to respond<br>I wouldn't know what to do. |
| | *Response based on need for more information* | • Conditional response based on need for more information | 23 | Depends on who it is. |
| | *Alternating different responses* | • Alternate between different responses | 1 | I would probably alternate between talking animatedly and going silent |
| **Non-Engagement** | *Engage someone else* | • Engage other people/someone else | 7 | I've done all I can do so I would speak to other people until they arrive |
| | *Focus on other activity* | • Refocus attention on something else or other activity<br>• Do something alone<br>• Leave the other person after an exchange<br>• Continue/resume doing what you were doing prior to the interaction | 89 | I would make alternative plans for myself.<br>I think I'd go and read a book in bed<br>I will leave them after saying goodbye<br>go back to watching tv |
| | *Limit interaction* | • Limit interaction with the other person | 8 | I would say hi to my friend and then would continue on my way. |
| | *Limit thinking about interaction* | • Not think much about situation | 4 | not think much of it. |
| | *No Response* | • Do nothing/stop doing what you were doing<br>• No response<br>• Wait for other person to do something | 26 | Nothing<br>Not respond at all.<br>Wait for a response |
| | *Not engaging scenario* | • Would not do what is in this scenario | 1 | Cant imagin—would never do this |

*Typological errors in the examples reflect the unedited qualitative data as it was entered into the experiment by participants

**Table 5. Observed percentages of participants that endorsed each qualitatively coded action across rejection conditions and their mean Inventory of Interpersonal Problem (IIP) t-score (standard deviation).**

| Action Category | IIP T-Score for Endorsing Participants (*SD*) | Total (%) | Control (%) | Ambiguous (%) | Rejection (%) |
|---|---|---|---|---|---|
| **Engaging** | 56.7 (9.4) | 93.0 | 84.2 | 62.7 | 31.0 |
| **Collaborating** | 56.7 (9.3) | 84.2 | 16.5 | 68.4 | 57.6 |
| **Supporting** | 56.9 (9.6) | 50.6 | 11.4 | 10.1 | 38.6 |
| **Attacking** | 59.0 (8.1) | 28.5 | 0.6 | 8.9 | 23.4 |
| **Avoiding** | 58.5 (8.6) | 65.2 | 5.7 | 38.6 | 48.7 |
| **Affect Expression** | 57.1 (9.7) | 59.5 | 43.7 | 20.9 | 15.2 |
| Positive | 56.4 (9.1) | 48.7 | 43.0 | 4.4 | 7.0 |
| Negative | 58.7 (11.0) | 27.2 | 1.3 | 18.4 | 8.9 |
| **Uncertainty** | 57.4 (7.6) | 15.2 | 2.5 | 2.5 | 13.9 |
| **Non-engagement** | 56.9 (9.7) | 69.6 | 31.0 | 29.7 | 38.6 |

to do in the afternoon, a negative comment would include responding to the romantic partner by saying "so you don't know me enough to plan things for us".

**Avoiding.** Avoiding responses included avoidance of different parts of the interaction, avoiding the situation and avoiding the person. Avoidance of the interaction included avoiding to share information about oneself (e.g., "I would make a note not to vent to that person again in future"), avoidance of expressing feelings or affect (e.g., "I would keep how I was feeling to myself") and avoidance of addressing any issues with the other person by "sulking" or ruminating (e.g., "I would dwell on this for a very, very long time").

**Affect expression.** The affect expression category included actions that displayed how the participant was feeling to the other person. This category was divided into two categories including expressions of positive and negative affect as they were considered to represent opposite sides of the affect spectrum. Positive expressions of affect including thanking or expressing liking, joking, expressions of physical affection and non-verbal expressions of positive affect expressions such as smiling and laughing (Table 4). Negative expressions of affect including general comments relating to negative emotional reactions such as "over-reacting", "letting all my emotions take over" or non-verbal gestures intended to communicate feelings such as sadness (e.g., "cry") and confusion (e.g., "show being puzzled about seeing her out and about").

**Uncertainty.** A minority of respondents described uncertainty about their actions or responses to the presented scenarios. Most of these responses included participants who reported different or conditional actions dependant on other factors to what was depicted in the scenario. Some examples of conditional responses included participants suggesting the action depended on other factors (e.g., "depends on who it is") and participants who indicated different responses based on additional information (e.g., "If I am involved we may be able to talk about it. If it isn't, they can leave the house."). Other types of responses coded in the uncertainty category included participants who indicated hesitation about how to respond, participants who alternated between two apparently opposite responses, and participants who indicated they were unsure about how to respond to the scenario (Table 4).

**Non-engagement.** The non-engagement category included actions where participants reported no response to the scenario or where participants engaged other parts of the scenario rather than engaging the person depicted in the vignette. The non-engagement category included responses that chose to engage other parts of the scenario than the subject of the scenario (e.g., "I would speak to other people until they arrive"), doing activities alone or resuming activities prior to the interaction (e.g., "I think I'd go and read a book in bed", "go back to watching tv") and not thinking about the situation in the vignette (e.g., "not think much of it"). The responses also included actions where participants chose to do nothing further to what was in the scenario, cease doing what they were doing, wait for a response from the other person or those that felt they would simply not find themselves in the scenario presented in the first place (Table 4). These responses were differentiated from avoidance responses as non-engagement followed an attempt to engage the other person first or where the response did not indicate overt avoidance. For example, in ambiguous vignette eight (S1 Appendix) where the participant's romantic partner is not home when they are ready to engage in an activity together, we did not consider the response "I would make alternative plans for myself" as an intention to avoid the romantic partner.

## Hypotheses 1: Degree of rejection on interpersonal responses

**Self-blame.** Significant differences in self-blame were found between the different vignette scenarios ($F(8, 1252) = 8.09$, $p < .001$, $f^2_m < .01$, $f^2_c = .02$) suggesting that it was appropriate to control for differences in vignettes. Participants reported increased levels of self-blame as the

degree of rejection in the interpersonal scenarios increased ($F(2, 1252) = 101.50$, $p < .001$, $f^2_m$ $= .01$, $f^2_c = .07$). Participants reported higher levels of self-blame in the ambiguous scenarios ($M = 21.37$, $SE = 1.24$, $t(1252) = -9.66$, $p < .001$) and rejection scenarios ($M = 26.94$, $SE = 1.24$, $t(1252) = -13.90$, $p < .001$) when compared to the control scenarios ($M = 8.72$, $SE = 1.24$). Participants also reported higher levels of self-blame in the rejection scenarios when compared to the ambiguous scenarios ($t(1252) = -4.25$, $p < .001$).

**Other-blame.** Significant differences in other-blame were found between the different vignette scenarios ($F(8, 1252) = 6.69$, $p < .001$, $f^2_m < .01$, $f^2_c = .01$) suggesting that it was appropriate to control for differences in vignettes. Participants reported increased levels of other-blame as the degree of rejection in the interpersonal scenarios increased ($F(2, 1252) = 198.41$, $p < .001$, $f^2_m = .04$, $f^2_c = .13$). Participants reported higher levels of other-blame in the ambiguous scenarios ($M = 29.98$, $SE = 1.47$, $t(1252) = -13.23$, $p < .001$) and rejection scenarios ($M = 39.67$, $SE = 1.47$, $t(1252) = -19.51$, $p < .001$) when compared to the control scenarios ($M = 9.57$, $SE = 1.47$). Participants also reported higher levels of other-blame in the rejection scenarios when compared to the ambiguous scenarios ($t(1252) = -6.29$, $p < .001$).

**Intentionality.** Significant differences in perceived intentionality were found between the different vignette scenarios ($F(8, 1252) = 17.22$, $p < .001$, $f^2_m < .01$, $f^2_c = .03$) suggesting that it was appropriate to control for differences in vignettes. Participants reported significant differences in levels of perceived intentionality across control, ambiguous and rejection conditions ($F(2, 1252) = 62.65$, $p < .001$, $f^2_m = .01$, $f^2_c = .04$). Participants reported higher levels of perceived intentionality in the control condition ($M = 54.51$, $SE = 1.87$) when compared to the ambiguous scenarios ($M = 32.86$, $SE = 1.87$, $t(1252) = 10.98$, $p < .001$) and rejection scenarios ($M = 47.39$, $SE = 1.87$, $t(1252) = 3.61$, $p < .001$). Participants also reported higher levels of perceived intentionality in the rejection scenarios when compared to the ambiguous scenarios ($t(1252) = -7.37$, $p < .001$).

**Stability.** Significant differences in perceived stability of actions were found between the different vignette scenarios ($F(8, 1252) = 4.56$, $p < .001$, $f^2_m < .01$, $f^2_c = .01$) suggesting that it was appropriate to control for differences in vignettes. Participants reported significant differences in levels of perceived stability of actions across control, ambiguous and rejection conditions ($F(2, 1252) = 60.98$, $p < .001$, $f^2_m = .01$, $f^2_c = .03$). Participants reported higher levels of perceived stability of actions in the control condition ($M = 51.51$, $SE = 1.68$) when compared to the ambiguous scenarios ($M = 31.83$, $SE = 1.68$, $t(1252) = 10.36$, $p < .001$) and rejection scenarios ($M = 35.37$, $SE = 1.68$, $t(1252) = 8.50$, $p < .001$). No significant differences were found between perceived stability of actions when comparing ambiguous and rejection scenarios ($t(1252) = -1.86$, $p = .063$).

**Globality.** Significant differences in perceived globality of actions were found between the different vignette scenarios ($F(8, 1252) = 5.85$, $p < .001$, $f^2_m < .01$, $f^2_c = .01$) suggesting that it was appropriate to control for differences in vignettes. Participants reported increased levels of perceived globality of actions as the degree of rejection in the interpersonal scenarios increased ($F(2, 1252) = 162.90$, $p < .001$, $f^2_m = .03$, $f^2_c = .12$). Participants reported higher levels of perceived globality of actions in the ambiguous scenarios ($M = 30.17$, $SE = 1.50$, $t(1252) = -11.78$, $p < .001$) and rejection scenarios ($M = 39.58$, $SE = 1.50$, $t(1252) = -17.73$, $p < .001$) when compared to the control scenarios ($M = 11.56$, $SE = 1.50$). Participants also reported higher levels of perceived globality of actions in the rejection scenarios when compared to the ambiguous scenarios ($t(1252) = -5.96$, $p < .001$).

**Emotional distress.** Significant differences in emotional distress were found between the different vignette scenarios ($F(8, 1252) = 14.77$, $p < .001$, $f^2_m < .01$, $f^2_c = .01$) suggesting that it was appropriate to control for differences in vignettes. Participants reported increased levels of emotional distress as the degree of rejection in the interpersonal scenarios increased ($F(2$,

1252) = 466.88, $p < .001$, $f^2_m = .13$, $f^2_c = .34$). Participants reported higher levels of emotional distress in the ambiguous scenarios ($M = 32.97$, $SE = 1.3$, $t(1252) = -21.13$, $p < .001$) and rejection scenarios ($M = 44.48$, $SE = 1.3$, $t(1252) = -29.68$, $p < .001$) when compared to the control scenarios ($M = 4.58$, $SE = 1.3$). Participants also reported higher levels of emotional distress in the rejection scenarios when compared to the ambiguous scenarios ($t(1252) = -8.56$, $p < .001$).

**Engaging.** Participants reported decreased odds of engaging the person in the scenario as the degree of rejection in the interpersonal scenarios increased ($\chi^2(2, N = 474) = 68.86$, $p < .001$, $f^2_m = .06$, $f^2_c = .09$). In particular, participants were 3.31 times less likely to engage in the ambiguous condition ($P = 0.63$, $SE = 0.04$, $OR = 3.31$, $z(474) = 4.23$, $p < .001$) and 13.10 times less likely to engage in the rejection condition ($P = 0.30$, $SE = 0.04$, $OR = 13.10$, $z(474) = 8.20$, $p < .001$) when compared to the control condition ($P = 0.85$, $SE = 0.03$). Participants were also 3.96 times less likely to engage in the rejection condition when compared to the ambiguous condition ($OR = 3.96$, $z(474) = 5.36$, $p < .001$).

**Collaborating.** Participants reported increased odds of collaborating with the other person about the issue as the degree of rejection in the interpersonal scenarios increased ($\chi^2(2, N = 474) = 66.62$, $p < .001$, $f^2_m = .07$, $f^2_c = .18$). In particular, participants were 15.38 times more likely to collaborate with the other person about the issue in the ambiguous condition ($P = 0.71$, $SE = 0.04$, $OR = 0.07$, $z(474) = -7.96$, $p < .001$) and 9.17 times more likely to collaborate in the rejection condition ($P = 0.59$, $SE = 0.04$, $OR = 0.11$, $z(474) = -6.85$, $p < .001$) when compared to the control condition ($P = 0.14$, $SE = 0.03$). No significant differences were found in the odds of collaborating in the rejection condition when compared to the ambiguous condition ($OR = 1.69$, $z(474) = 2.01$, $p = .045$).

**Supporting.** Participants reported increased odds of supporting as the degree of rejection in the interpersonal scenarios increased ($\chi^2(2, N = 474) = 41.00$, $p < .001$, $f^2_m = .03$, $f^2_c = .04$). No differences were observed in supporting between the control ($P = 0.11$, $SE = 0.03$) and ambiguous conditions ($P = 0.10$, $SE = 0.03$, $OR = 1.17$, $z(474) = 0.42$, $p = .678$). However, participants were 5.05 times more likely to support the other person in the rejection condition when compared to the control condition ($P = 0.38$, $SE = 0.04$, $OR = 0.20$, $z(474) = -5.14$, $p < .001$) and 5.88 times more likely to support in the rejection condition when compared to the ambiguous condition ($OR = 0.17$, $z(474) = -5.36$, $p < .001$).

**Fighting.** Participants reported increased odds of fighting as the degree of rejection in the interpersonal scenarios increased ($\chi^2(2, N = 474) = 21.12$, $p < .001$, $f^2_m = .17$, $f^2_c = .50$). In particular, participants were 76.92 times more likely to fight in the rejection condition ($P = 0.17$, $SE = 0.05$, $OR = 0.01$, $z(474) = -3.59$, $p < .001$) when compared to the control condition ($P = 0.003$, $SE = 0.004$). Participants were 3.79 times more likely to fight in the rejection condition when compared to the ambiguous condition ($P = 0.05$, $SE = 0.02$, $OR = 0.26$, $z(474) = -3.43$, $p < .001$). No significant differences were found in the odds of fighting in the control condition when compared to the ambiguous condition ($OR = 0.05$, $z(474) = -2.49$, $p = .013$).

**Avoiding.** Participants reported increased odds of avoiding as the degree of rejection in the interpersonal scenarios increased ($\chi^2(2, N = 474) = 50.87$, $p < .001$, $f^2_m = .11$, $f^2_c = .22$). In particular, participants were 12.5 times more likely to avoid in the ambiguous condition ($P = 0.37$, $SE = 0.05$, $OR = 0.08$, $z(474) = -6.14$, $p < .001$) and 20.41 times more likely to avoid in the rejection condition ($P = 0.49$, $SE = 0.05$, $OR = 0.05$, $z(474) = -7.11$, $p < .001$) when compared to the control condition ($P = 0.04$, $SE = 0.02$). No significant differences were found in the odds of avoiding in the rejection condition when compared to the ambiguous condition ($OR = 0.61$, $z(474) = -1.94$ $p = .053$).

**Expressing positive affect.** Participants reported decreased odds of expressing positive affect as the degree of rejection in the interpersonal scenarios increased ($\chi^2(2, N = 474) = 57.89$, $p < .001$, $f^2_m = .11$, $f^2_c = .13$). In particular, participants were 20.91 times less likely to

express positive affect in the ambiguous condition ($P = 0.04$, $SE = 0.02$, $OR = 20.91$, $z(474) =$ 5.99, $p < .001$) and 11.65 times less likely to express positive affect in the rejection condition ($P = 0.06$, $SE = 0.02$, $OR = 11.65$, $z(474) = 6.01$, $p < .001$) when compared to the control condition ($P = 0.43$, $SE = 0.04$). No differences were found in the odds of expressing positive affect in the ambiguous condition when compared to the rejection condition ($OR = 0.56$, $z(474) =$ -1.03, $p = .303$).

**Expressing negative affect.** Participants reported significant differences in the odds of expressing negative affect across the rejection conditions in the interpersonal scenarios ($\chi^2(2, N = 474) = 15.79$, $p < .001$, $f^2_m = .10$, $f^2_c = .10$). In particular, participants were 23.26 times more likely to express negative affect in the ambiguous condition ($P = 0.18$, $SE = 0.03$, $OR = 0.04$, $z(474) = -3.41$, $p < .001$) when compared to the control condition ($P = 0.01$, $SE = 0.01$). No differences were found in the odds of expressing negative affect when comparing the rejection condition ($P = 0.08$, $SE = 0.02$) to the control condition ($OR = 0.11$, $SE = 0.10$, $z(474) = 2.38$, $p = .018$) and the ambiguous condition ($OR = 2.46$, $SE = 0.90$, $z(474) = 2.60$, $p = .014$).

**Uncertainty.** The odds of expressing uncertainty about actions was not assessed due to extremely low base rates and low estimated marginal probabilities of uncertainty across the control ($P < 0.01$), ambiguous ($P < 0.01$) and rejection conditions ($P < 0.01$).

**Non-engagement.** No differences in the odds of non-engagement were found across the control ($P = 0.31$), ambiguous ($P = 0.29$) and rejection conditions ($P = 0.38$, $\chi^2(2, N = 474) = 3.35$, $p = .187$, $f^2_m < .01$, $f^2_c < .01$).

### Hypothesis 2: Moderation of EMSs on the impact of rejection on interpersonal responses

**Perceived rejection.** Participants with higher EMS scores also reported higher levels of perceived rejection overall ($F(1, 156) = 34.34$, $p < .001$, $f^2_m = .03$, $f^2_c < .01$). However, this effect was not statistically significant after adding the cross-level interaction between degree of rejection and EMS scores ($ß = 0.03$, $t(156) = 1.10$, $p = .272$, $f^2_m = .01$, $f^2_c = .02$). EMSs significantly moderated the relationship between degree of scenario rejection and perceived rejection ($F(2, 1252) = 18.43$, $p < .001$) such that participants with higher EMS scores reported higher levels of perceived rejection as the degree of scenario rejection increased. Differences in perceived rejection and cognitive responses for people with high, low and mean levels of EMSs across rejection conditions are displayed in Fig 2.

**Self-blame.** Participants with higher EMS scores also reported higher levels of self-blame overall ($F(1, 156) = 53.3$, $p < .001$, $f^2_m = .03$, $f^2_c < .01$). EMSs significantly moderated the relationship between degree of scenario rejection and self-blame ($F(2, 1252) = 14.0$, $p < .001$, $f^2_m = .01$, $f^2_c = .01$) such that participants with higher EMS scores reported higher levels of self-blame as the degree of scenario rejection increased.

**Other-blame, intentionality, stability and globality.** Participants with higher EMS scores reported higher levels of other-blame ($F(1, 156) = 13.16$, $p < .001$, $f^2_m = .01$, $f^2_c < .01$) and attributed the other person's actions to be more intentional ($F(1, 156) = 9.23$, $p = .003$, $f^2_m < .01$, $f^2_c < .01$), stable ($F(1, 156) = 25.22$, $p < .001$, $f^2_m = .01$, $f^2_c < .01$) and to have a more global effect on the relationship irrespective of rejection condition ($F(1, 156) = 19.55$, $p < .001$, $f^2_m = .01$, $f^2_c < .01$). However, EMSs did not significantly moderate the relationship between degree of scenario rejection and other-blame, intentionality, stability or globality (Table 1).

**Emotional distress.** Participants with higher EMS scores also reported higher levels of emotional distress ($F(1, 156) = 24.85$, $p < .001$, $f^2_m = .03$, $f^2_c < .01$). However, this effect became non-significant after adding the cross-level interaction between degree of rejection and EMS scores ($ß = 0.02$, $t(156) = 0.03$, $p = .258$). EMSs significantly moderated the relationship between degree of scenario rejection and emotional distress ($F(2, 1252) = 12.64$, $p < .001$,

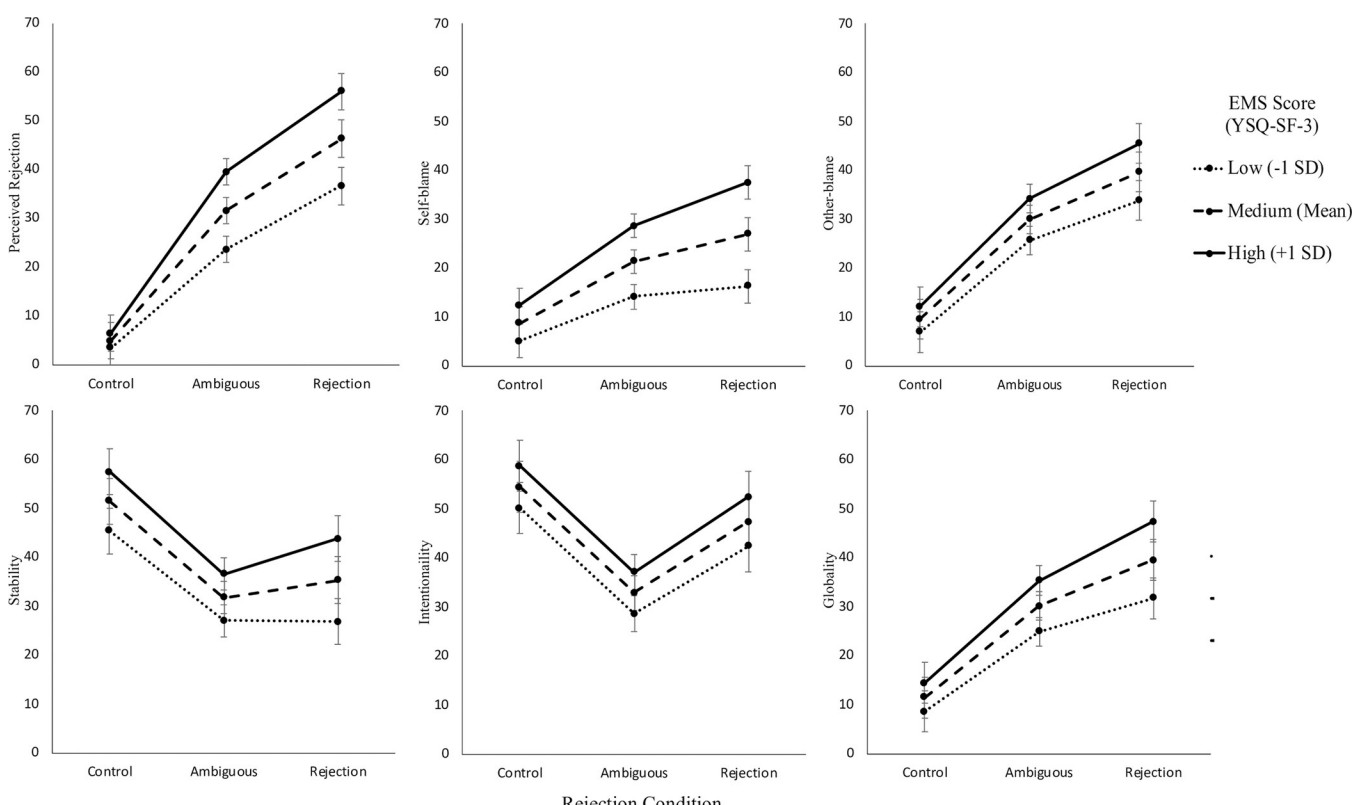

**Fig 2. Mean cognitive responses across conditions of interpersonal scenario rejection scored between 0 (*Not at all rejected; Extremely unlikely*) to 100 (*Extremely rejected; Extremely likely*).**

$f^2_m = .01, f^2_c = .01$) such that participants with higher EMS scores reported higher levels of emotional distress as the degree of scenario rejection increased. Differences in emotional distress for people with high, low and mean levels of EMSs across rejection conditions are displayed in Fig 3.

**Behavioural responses.** Participant EMS scores did not predict the odds of any qualitatively coded action and did not moderate the relationship between degree of scenario rejection and the odds of any qualitatively coded action (Table 2). Differences in behavioural responses for people with high, low and mean levels of EMSs across rejection conditions are displayed in Fig 4.

## Discussion

The aim of the present study was to assess if EMSs moderate the impact of interpersonal situations on interpersonal responses including negative cognitions, distress and unhelpful behavioural responses. To assess these aims, we developed a novel repeated measures experiment using written interpersonal scenarios that manipulated the degree of rejection by assessing participants across control (acceptance), ambiguous and rejection conditions as a means of manipulating the interpersonal situation. The moderation of EMSs was assessed by comparing changes in a participant's interpersonal responses in rejection conditions across participants with varying levels of EMS endorsement. Rejection manipulations across scenarios were supported by the data with participants reporting increasing levels of perceived rejection as the degree of rejection increased across conditions. Perceived rejection scores in the experiment

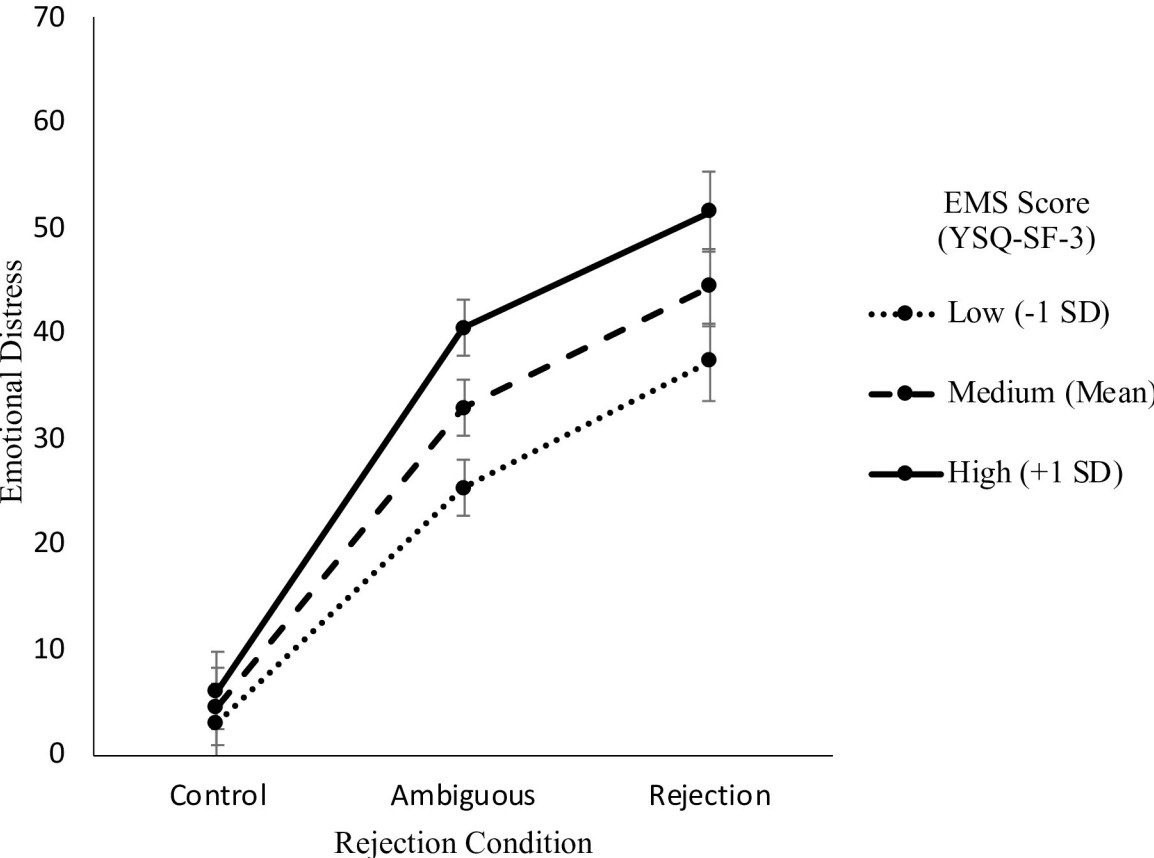

**Fig 3. Mean emotional distress across conditions of interpersonal scenario rejection scored between 0 (*Not at all upset or angry*) to 100 (*Extremely upset or angry*).**

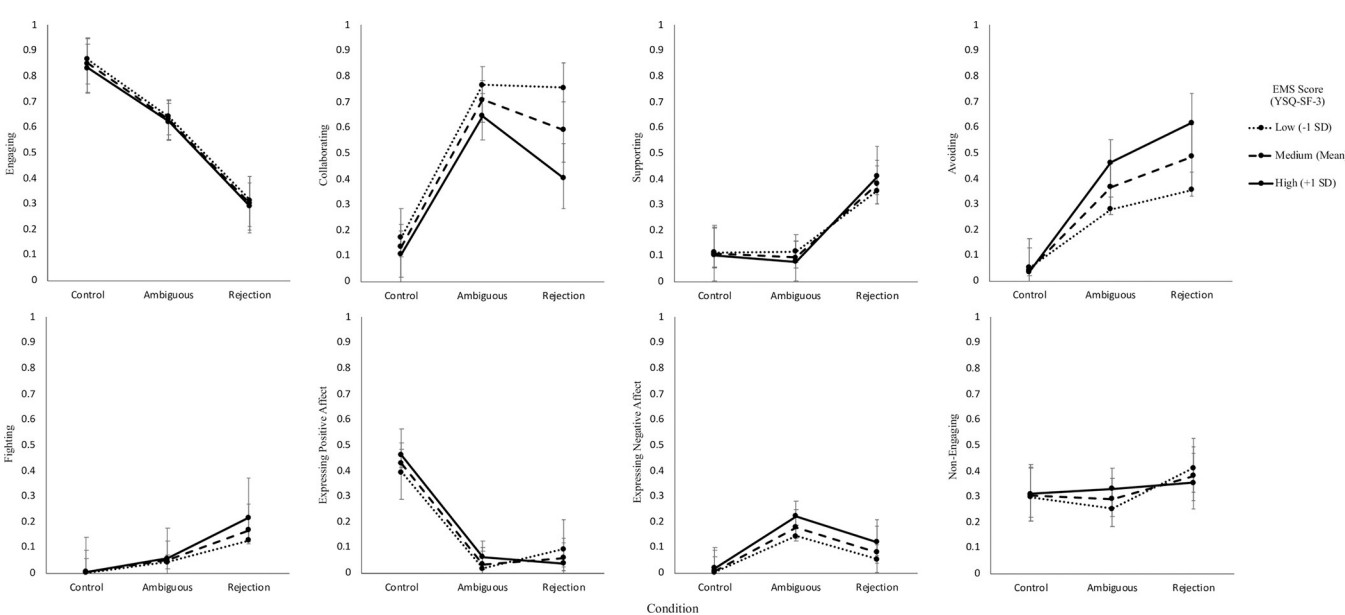

**Fig 4. Probabilities of engaging in different types of actions by rejection condition and EMS score.**

were also found to have moderate positive correlations with similar measures including interpersonal problems, rejection sensitivity and symptoms of depression, anxiety and stress. Together these findings supported the validity of our experimental paradigm and its manipulation of interpersonal rejection.

Negative cognitive attributions in response to interpersonal scenarios in the experiment and EMSs were also found to have small to high positive associations with trait measures of problematic interpersonal behaviours, rejection sensitivity and psychiatric symptoms. These findings are consistent with previous research finding individuals who make internal, global and stable attributions of negative events also experience more negative outcomes including depressive symptoms, reduced performance and hostility [64, 65]. These findings were also consistent with recent meta-analyses finding moderate to high associations between EMSs, interpersonal problems and psychiatric symptoms [22, 66].

## Interpersonal responses to rejection

To assess participant responses to the interpersonal scenarios, participants were asked to rate the likelihood of negative cognitive attributions, self-reported emotional distress and provide written responses to how participants might respond to the scenarios across conditions. It was predicted that participants would show increased negative cognitions, emotional distress and differences in behaviour as the degree of rejection increased. In support of the present study's hypotheses, participants were found to report higher levels of emotional distress as the degree of rejection increased. These findings are consistent with previous research reporting increased negative affect following interpersonal rejection [67, 68].

We found that in general participants also reported increased negative cognitions as the degree of rejection increased. In particular, participants reported higher levels of self-blame, other-blame and predicted higher likelihood that the issues in the scenario would globally impact other parts of their relationship as the degree of rejection increased. These findings are consistent with past research showing that people tend to report lower self-esteem and increased other-blame following rejection [69, 70]. Interestingly, participants reported stronger convictions that the person in the scenario was likely to change their behaviour (low stability) in the ambiguous and rejection conditions when compared to the control condition. Participants also perceived behaviour to be more intentional in the control and rejection conditions when compared to the ambiguous condition. However, participants perceived actions in the control condition to have a higher likelihood of being intentional than in the rejection condition. Together, these findings suggest that participants reported lower convictions in perceived intent and stability of rejection in the current experiment. This finding opposes previous experimental research where participants reported increased perceptions of hostile intent following rejection in children [65]. Unlike the experiment conducted by Reijntjes et al. [65] where rejection occurred from peers not known to the participants, in this study rejection was presented as a single transgression within a close interpersonal relationship (i.e., from a family member, friend or romantic partner). Fletcher and Kerr [71] found that individuals were more likely to make positive attribution biases about a person's actions and transgressions in relationships where there was greater intimacy. Therefore, if participants held relatively strong relationships with their loved ones, it is likely that they might attribute situations involving acceptance to be intentional and stable and unlikely that they would attribute the rejection to be stable or intentional following a single transgression. In contrast, it remains possible that in relationships with limited intimacy (e.g., strangers or peers unknown to the participant) participants might respond more negatively following rejection than how they responded in the present study [65].

Using inductive content analysis of the written responses to the interpersonal scenarios, the present study identified eight overall categories of behavioural responses. One such response included engaging the other person which comprised of actions such as approaching, greeting, making conversation or planning to spend time with the person in the scenario. Unsurprisingly, participants were less likely to engage the person as the degree of rejection increased. Alternatively, participant responses suggested increasing odds of attempting to collaborate about the issue or avoid the person in the scenario in the ambiguous and rejection conditions and increased odds of fighting in the rejection condition. Responses coded as collaborating generally reflected a desire to communicate about the issues presented in the scenarios. In contrast, responses coded in the avoiding or fighting categories consisted of avoiding parts of the interaction, the situation or even the person or engaging in behaviours such as confronting and arguing, making negative comments or retaliating with an intent to hurt the other person. These findings are consistent with previous research that found that increased communication and avoidance following experiences of rejection [67]. These findings are also consistent with research supporting increased aggression following instances of rejection [65].

The present study's qualitative analysis also identified supporting behavioural responses. Supporting responses generally prioritised the needs of the person in the scenario or involved sacrificing the participant's interpersonal goals. Participants reported comparably low levels of supporting actions in the control and ambiguous scenarios but significant increases in supporting in the rejection condition. This finding is surprising considering research generally suggests that people tend to show reduced pro-social behaviour following rejection [72]. However, people with low social anxiety have been found to engage in increased pro-social behaviour following rejection while people high in social anxiety did not suggesting some interpersonal characteristics might promote supportive behaviours following rejection [73]. It is also possible that external attributions such as perceptions that your loved one is acting rejecting due to a personal struggle might motivate supporting behaviour after being rejected by someone you typically share a close relationship with. Alternatively, given acceptance was also classified as a supporting strategy it is also possible that acceptance might be more common in situations involving high rejection than in situations where rejection is ambiguous.

The qualitative data also indicated that some participants engaged in behaviours which reflected emotional responses such as expressions of positive and negative affect. Unsurprisingly most expressions of positive affect occurred in the control condition with significantly lower odds of expressing positive affect in both the ambiguous and rejection conditions. In contrast, expressions of negative affect were found to be higher in the ambiguous condition when compared to the control condition. While this finding is consistent with low levels of emotional distress reported in the control condition, participants expressed no differences in negative affect between the control and rejection conditions despite participants reporting the highest amount of subjective distress in the rejection condition. Participants might have reported fewer expressions of negative affect in the qualitative responses because open ended questions did not specifically ask about emotional responses but rather how a participant would act in response to the scenario. Alternatively, increased avoidance in the rejection condition could lead some participants to suppress or hide any expressions of negative affect at higher levels of rejection. Some researchers have argued that avoidance strategies following interaction may at times function to avoid further rejection or damage to the relationship [67]. A minority of behavioural responses were also categorised as uncertainty about how to act or not engaging an interpersonal response. However, changes in uncertainty were not assessed across conditions due to very low base rates and there were no significant differences in responses that did not engage an interpersonal response across any of the experiment's conditions. Together, our findings combined quantitative and qualitative data to provide researchers

with a novel experimental paradigm for assessing the relationship between rejection and emotional, cognitive and behavioural responses to allow for stronger causal attributions in the study of interpersonal behaviour. The categories of engaging, supporting, collaborating, avoiding and fighting also share similarities with the schema coping styles of avoidance, overcompensation and surrender suggested by Young, Klosko and Weishaar [7] and Polyvagal Theory [74] where feelings of safety and security are argued to promote behaviour associated with connecting, while increased stress promotes activation of fight and flight responses.

## The moderating effect of EMSs on interpersonal responses to rejection

To assess the moderating effect of EMSs on the relationship between rejection and interpersonal responses. We predicted that individuals with higher EMSs would report increased negative cognitions, distress and problematic behaviours as the degree of rejection increased. Participants reporting higher EMSs were found to report significantly higher levels of perceived rejection, emotional distress and negative attributions (including self-blame, other-blame, globality, stability and intentionality) across all conditions. However, EMSs only moderated the relationship between rejection condition and perceived rejection, emotional distress and self-blame. While these findings provide support for the Schema Therapy model, which argues that individuals with EMSs show increased levels of distress and negative cognitions when EMSs are activated by stressful interpersonal contexts (e.g., rejection), it is surprising that EMSs only moderated the effect of rejection on self-blame and perceived rejection. Together these findings suggest that individuals with higher EMSs tend to report greater distress and negative cognitions regardless of the interpersonal context or situation. Given the finding that interpersonal rejection only activated select negative cognitions in people with higher EMSs, it is possible that EMSs might only moderate the relationship between interpersonal contexts and cognitions unique to that individual's schema (e.g., only cognitions related to abandonment might be activated in people with abandonment schema). Further research could help clarify these findings by assessing the moderating effect of EMSs on cognitions and scenarios congruent to a specific EMS rather than EMSs in general.

Contrary to predictions, individuals reporting higher EMSs did not show any changes in the odds of engaging in any of the categories of behavioural responses. Changes in the odds of engaging in different behavioural responses were also not observed for EMSs scores across rejection conditions, providing no support for a moderating effect of EMSs on the relationship between rejection and behavioural responses to interpersonal scenarios. These findings suggested that EMSs alone may be less likely to predict interpersonal behaviour. These findings are in contrast to the evidence supporting a link between EMSs and problematic interpersonal behaviour, such as past research that has found individuals with EMSs to report higher levels of aggression [14, 15], interpersonal conflict [17], and trait problem interpersonal behaviour patterns [1, 21]. One explanation for these findings is that most of the studies to date have measured trait dispositional aggression or self-reported perceptions of conflict and interpersonal problems. Given individuals with EMSs by definition hold more negative views of themselves and report higher levels of self-blame, it is possible that use of self-report measures might over-estimate the degree of problematic behaviour in people with higher EMSs [22]. However, adolescents with EMSs have also been assessed as more aggressive from reports from their peers and mothers in two separate studies [75, 76]. While limited, previous research suggests that it is likely that individuals with EMSs engage in more unhelpful interpersonal behaviour. Consequently, it is possible that factors outside of the scope of the present study might influence whether a person with EMSs engages in unhelpful interpersonal behaviour. For example, researchers have found hostility attributions to mediate the relationship between peer rejection and aggressive behaviour [65].

## Theoretical and clinical implications

The finding that EMSs predicted and moderated the impact of interpersonal rejection on cognitive and emotional interpersonal responses suggests that there is utility in addressing EMSs in therapy to reduce convictions in negative cognitions and emotional distress in interpersonal relationships. These findings are supported by previous research finding reductions in emotional distress and negative cognitions in select disorders such as major depressive disorder and borderline personality disorder following Schema Therapy and following a couples based schema therapy intervention [77–79]. However, in contrast to the position of the schema therapy model that argues negative cognitions and emotional distress are likely to be activated in stressful situations or when the context is congruent with a schema [7], we found people with higher EMSs were likely to show higher negative cognitions and emotional distress overall regardless of the interpersonal situation.

The findings from the present study also suggest that interpersonal situation or degree of rejection was more likely to predict interpersonal behaviour than EMSs given EMSs were not found to predict behaviour responses. These findings suggest that interpersonal behaviour might be more challenging to change without considering broader interpersonal contextual factors. Such contextual factors might include dyadic interactions that are best addressed using interventions such as couples therapy [80]. For example, Fals-Stewart and Clinton-Sherrod [80] found couples therapy to be superior to individual therapy in reducing intimate partner violence in a sample of offending males. Interventions might also need to consider increasing awareness of interpersonal contexts such as learning to identify when rejection is ambiguous as participants were more likely to collaborate to resolve perceived issues in the ambiguous rejection scenarios while reporting increased likelihood of fighting and supporting the other person when rejection was perceived as less ambiguous.

## Limitations and considerations for future research

Together the findings of the present study provide support for the theory that people with higher EMSs experience higher levels of emotional distress and negative cognitions when presented with higher degrees of interpersonal rejection and that EMSs moderate the relationship between the emotional and cognitive responses to interpersonal rejection. However, there are some limitations to consider with respect to the findings of the present study. The limited differences observed in behaviours across people with EMSs might in part reflect the reduced variability of actions due to having action categories as binary outcome variables restricted to select single incident rejection scenarios. In particular, actions were split into different categories and each individual's action was coded as if the response reflected the category or not. It is possible that people with EMSs might simply express behavioural responses at higher levels of intensity or at greater frequencies; characteristics unlikely to be detected using binary coding of presence of type of behaviour (or not) or single scenario experiences of rejection.

This study also assessed EMSs as an overall EMS score not allowing for different EMSs to be assessed individually. It is likely that changes in responses across interpersonal situations depicting rejection might be more apparent in EMSs more specific to rejection and not others [22]. For example, the disconnection and rejection EMS domain subscale which was not used in the present study due to poor factor structure might show stronger associations between interpersonal rejection and responses. In addition, it is possible that specific EMSs or other individual differences may lend themselves more specifically to certain styles of coping or behavioural actions. For example, researchers have predicted and found the EMS of insufficient self-control and discipline to be more strongly associated with aggressive behaviour [39].

This study also embedded responses to different types of relationships within its design by including an equal number of scenarios specific to friends, family and romantic partners across conditions. While this was intended to control for differences across close relationships, we were unable to assess for differences across types of relationships due to the variability in responses between individual scenarios. It should also be noted that while we were able to control for scenario variation in our statistical analyses for cognitive and emotional responses, the number of categorical variables when analysing our binary behavioural response variables created non-convergence issues when fitting our logistic regressions, preventing us from controlling for scenario variation when assessing the impact of condition and EMSs on behavioural responses. Given that our study has identified types of behavioural responses to the scenarios used, future research might benefit from using such results or other established models of interpersonal behaviour to guide the assessment of behavioural responses. This could include quantitative measures such as likelihood assessments of different responses, frequency count measures or measures of significant others' cognitions, emotions and responses.

## Conclusion

Despite the above limitations, the findings of the present study are some of the first to show that EMSs can moderate the impact of interpersonal context on emotional and cognitive responses to those contexts. Our findings also suggest that people with EMSs hold higher levels of emotional distress and negative attributions about interpersonal situations in general, regardless of the interpersonal contexts (e.g., rejection). In some cases, this distress and negative attributions can then be worsened in stressful interpersonal situations (e.g., rejection). While we did not find any differences in behavioural responses between varying levels of EMSs, our study provides a comprehensive overview of the wide range of responses participants can display following rejection, which were organised into a set of categories. Most noteworthy types of responses included collaborating or engaging others, which occurred more frequently in the acceptance and ambiguous rejection conditions, while supporting, avoiding or fighting with the other person increased in the rejection condition. The present study also provides researchers with a novel experimental paradigm to strengthen causal inferences for the impact of rejection and activation of EMSs on emotional, cognitive and behavioural responses to rejection.

## Supporting information

**S1 Appendix. Interpersonal vignettes.** Friend, family and romantic vignettes developed for the study with variations based on acceptance, ambiguous and unambiguous rejection conditions.
(PDF)

**S2 Appendix. Example interpersonal vignette and response layout.** Example layout for presentation of interpersonal vignettes and responses to vignettes including perceived rejection, cognitions, emotional distress and qualitative behavioural responses.
(PDF)

**S1 Dataset. Qualitative coding of actions.** Inductive content analysis including all responses, codes and categories for the qualitatively coded behavioural responses.
(XLSX)

**S2 Dataset. Quantitative dataset.** Dataset used for statistical analysis for quantitative responses including perceived rejection, cognitions and emotional distress.
(CSV)

**S3 Dataset. Qualitative dataset.** Dataset used for statistical analysis for binary coded qualitative behavioural responses.
(CSV)

## Author Contributions

**Conceptualization:** Thomas Janovsky, Gavin I. Clark.

**Data curation:** Thomas Janovsky.

**Formal analysis:** Thomas Janovsky, Valerie Polad, Suzanne Cosh.

**Funding acquisition:** Thomas Janovsky.

**Investigation:** Thomas Janovsky.

**Methodology:** Thomas Janovsky, Gavin I. Clark.

**Project administration:** Thomas Janovsky.

**Resources:** Thomas Janovsky.

**Supervision:** Adam J. Rock, Einar B. Thorsteinsson, Gavin I. Clark.

**Writing – original draft:** Thomas Janovsky.

**Writing – review & editing:** Thomas Janovsky, Adam J. Rock, Einar B. Thorsteinsson, Valerie Polad, Suzanne Cosh.

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
