## [Decision Letter · Decision Letter 0]

12 Jan 2023

PONE-D-22-32865Assessing the relationship between early maladaptive schemas and interpersonal problems using interpersonal scenarios depicting rejectionPLOS ONE

Dear Dr. Janovsky,

Thank you for submitting your manuscript to PLOS ONE. After careful consideration, we feel that it has merit but does not fully meet PLOS ONE’s publication criteria as it currently stands. Therefore, we invite you to submit a revised version of the manuscript that addresses the points raised during the review process.

We look forward to receiving your revised manuscript.

Kind regards,

Supat Chupradit, Ph.D., M.Ed., B.Sc.(OT), B.P.A., B.Ed., B.A.

Academic Editor

PLOS ONE

Journal Requirements:

2. Please ensure that you have specified (1) whether consent was informed and (2) what type you obtained (for instance, written or verbal, and if verbal, how it was documented and witnessed). If your study included minors, state whether you obtained consent from parents or guardians. If the need for consent was waived by the ethics committee, please include this information.

Reviewers' comments:

Reviewer's Responses to Questions

**Comments to the Author**

1. Is the manuscript technically sound, and do the data support the conclusions?

Reviewer #1: Yes

Reviewer #2: Yes

2. Has the statistical analysis been performed appropriately and rigorously? 

Reviewer #1: Yes

Reviewer #2: Yes

3. Have the authors made all data underlying the findings in their manuscript fully available?

Reviewer #1: Yes

Reviewer #2: Yes

4. Is the manuscript presented in an intelligible fashion and written in standard English?

Reviewer #1: Yes

Reviewer #2: No

5. Review Comments to the Author

Reviewer #1: 1-Before analyzed the data the authors should check the assumption of multiple regression for instant normality, data distribution(linear or not ?), VIF, etc.

2-The result found between low and high level correlation and found statistically significant, therefore, the authors should describe in dissusion part (explain why and how ? )

Reviewer #2: 1. Abstract: Include the specific statistical analysis used in the investigation. Provide statistical results as well.

2. I think that Introduction provided a comprehensive setting of the context related to the issue addressed by the investigation. However, I found it to be quite lengthy for a journal research article, and seems more attuned to a thesis or dissertation. Some information within each subsection is redundant. I suggest for the authors to significantly reduce this section.

3. A similar observation with the subsection on the study aims. You may want to be explicit and succinct in reporting, instead of providing context.

4. Include an ethics statement at the start of the methods section.

5. The methods section is quite lengthy and may benefit from editing. There are times when you explain things, instead of just reporting explicitly. I suggest for the author to consult the STROBE checklist for the minimum information that should be reported in a correlation study such as this.

6. Change the subsection title power analysis to reflect sample size instead of the statistical process.

7. To clarify, your recruitment was not age group sensitive, hence older age participants who are enrolled in the 1st year psychology program were sampled? What was your inclusion criteria to begin with? Were there any race/nationality/ethnicity variances?

8. Include description of scores and how are they typically interpreted for your instruments

9. Clarify which among the instruments were administered electronically. Did you at the very least compute for any form of psychometric properties of the electronic version of instrument you have used?

10. I think you may need to have a separate subsection to fully describe the experimental paradigm (i.e., development, technical and electronic requirements, overall procedures).

11. Did you use the generic Jamovi software or did you include a specific statistical module? If you did the latter, do you have evidence to support the robustness of that module in carrying out the intended statistical tests?

12. How different are the text results from those reported in the tables? Maybe highlight the most significant or interesting findings in the text, because as it is right now, this section 9as with the rest of the submitted manuscript) is quite text-heavy.

11. In your discussion, instead of using the hypotheses as subsection titles, you may want to rethink of a more descriptive title encompassing of what is discussed in the respective subsections.

12. Were there any interesting or novel findings not found in the current evidence? Or how does your result differ among the studies your found?

6. PLOS authors have the option to publish the peer review history of their article (what does this mean?). If published, this will include your full peer review and any attached files.

Reviewer #1: No

Reviewer #2: No

---

## [Author Response · Author response to Decision Letter 0]

8 Feb 2023

Regarding: PONE-D-22-32865

Assessing the relationship between early maladaptive schemas and interpersonal problems using interpersonal scenarios depicting rejection

Dear Professor Supat Chupradit, 

Thank you for the opportunity to revise our manuscript and thank you to all the reviewers for their input and feedback. Please find our replies to the academic editor and reviewers in the comments below. 

Sincerely, 

Thomas Janovsky

School of Psychology

University of New England

Armidale NSW 2351

Australia

 

Response to Academic Editor

Journal Requirements:

REPLY: Manuscript and files have now been changed to reflect the PLOS One style templates (See track changes).

2. Please ensure that you have specified (1) whether consent was informed and (2) what type you obtained (for instance, written or verbal, and if verbal, how it was documented and witnessed). If your study included minors, state whether you obtained consent from parents or guardians. If the need for consent was waived by the ethics committee, please include this information.

REPLY: The following statement has been added to the beginning of the methods section of the paper: “This research was conducted with approval from the University of New England Human Research Ethics Committee (Approval Number: HE18-248) with all participants providing informed and signed written consent for participation.”

REPLY: We have now added the Australian Government in the Funding Information box. There is no award number for this source of funding but we have provided the Award name: Australian Postgraduate Award. 

REPLY: The funding provided for the following research project was a Higher Research Training Scholarship funded by the Australian Government. This training scholarship is provided to doctoral students to support the completion of a PhD. Given this scholarship is awarded to support living arrangements while completing a PhD, no grant number was provided when the scholarship was approved. 

REPLY: Author affiliations have been edited and updated in the manuscript to reflect the PLOS One formatting sample (see track changes)

REPLY: The following statement has been added to the beginning of the methods section of the paper: “This research was conducted with approval from the University of New England Human Research Ethics Committee (Approval Number: HE18-248) with all participants providing informed and signed written consent for participation.”

REPLY: The reference list has been reviewed to ensure it is correct (see track changes). References that have been removed have been removed due to significant reductions in text as recommended by Reviewer 2 (i.e. the references are no longer cited in the article). New references cited have now been added to the reference list. 

 

Response to Reviewers

Reviewer #1: 

1-Before analyzed the data the authors should check the assumption of multiple regression for instant normality, data distribution(linear or not ?), VIF, etc.

REPLY: The following paragraph has been added to clarify the process of checking the assumptions and decision making: “The assumptions of normality and linearity were satisfied in addition to model variables not showing any signs of multicollinearity. Visual inspection of the scatterplot of residuals suggested signs of double outward box heteroscedasticity which was confirmed by significant loglikelihood test differences when comparing the random-intercept only model of the absolute residuals to the full model of absolute residuals regressed on all of the predictors (Huang et al., 2022). Such heteroscedasticity is common in experiments with control conditions and likely reflected reduced variability of responses in the control scenarios when compared to the ambiguous and rejection scenarios (Korendijk et al., 2008). We decided to proceed with interpretation of the multilevel models despite the observed heteroscedasticity to preserve simplicity and interpretability of data given previous simulation studies have found multi-level models to be robust to violations in homoscedasticity when there are equal observations per condition (Huang et al., 2022; Korendijk et al., 2008)”

2-The result found between low and high level correlation and found statistically significant, therefore, the authors should describe in dissusion part (explain why and how ? )

REPLY: We have added the following paragraph to the discussion section to help describe the low and high correlations observed in the results: 

“Perceived rejection scores in the experiment were also found to have moderate positive correlations with similar measures including interpersonal problems, rejection sensitivity and symptoms of depression, anxiety and stress. Together these findings supported the validity of our experimental paradigm and its manipulation of interpersonal rejection.

Negative cognitive attributions in response to interpersonal scenarios in the experiment and EMSs were found to have small to high positive associations with trait measures of problematic interpersonal behaviours, rejection sensitivity and psychiatric symptoms. These findings are consistent with previous research finding individuals who make internal, global and stable attributions of negative events also experience more negative outcomes including depressive symptoms, reduced performance and hostility (Houston, 2016; Reijntjes et al., 2011). These findings were also consistent with recent meta-analyses finding moderate to high associations between EMSs, interpersonal problems and psychiatric symptoms (Bishop et al., 2022; Janovsky et al., 2020).”

Reviewer #2: 

1. Abstract: Include the specific statistical analysis used in the investigation. Provide statistical results as well.

REPLY: The Abstract has been substantially changed to allow for additional information regarding the statistical analysis and reporting of statistical results as follows: 

“Background: Early maladaptive schemas (EMSs) have been theorised to contribute to reoccurring interpersonal problems. Method: This study developed a novel experimental paradigm that aimed to assess if EMSs moderate the impact of interpersonal situations on interpersonal responses by manipulating the degree of rejection in a series of interpersonal vignettes depicting acceptance, ambiguous rejection and rejection. In a sample of 158 first-year psychology students (27.2% male; 72.2% female; 0.6% other) participant responses to interpersonal scenarios were measured including degree of perceived rejection, emotional distress, conviction in varying cognitive appraisals consistent with attribution theory and behavioural responses to scenarios. Qualitative data was analysed using inductive content analysis and statistical analyses were conducted using multi-level mixed effect linear and logistic regression models using the software Jamovi (Version 1.6, Jamovi Project, 2021). Results: People reporting higher EMSs reported increased emotional distress (F(1, 156) = 24.85, p < .001), perceptions of rejection (F(1, 156) = 34.33, p < .001), self-blame (F(1, 156) = 53.25, p < .001), other-blame (F(1, 156) = 13.16, p < .001) and more intentional (F(1, 156) = 9.24, p = .003), stable (F(1, 156) = 25.22, p < .001) and global (F(1, 156) = 19.55, p < .001) attributions but no differences in reported behavioural responses. The results also supported that EMSs moderate the relationship between interpersonal rejection and perceptions of rejection (F(2, 1252) = 18.43, p < .001), emotional distress (F(2, 1252) = 12.64, p < .001) and self-blame (F(2, 1252) = 14.00, p < .001). Conclusion: Together these findings argue that people with EMSs experience increased distress and select negative cognitions in situations where there are higher levels of rejection but that distress and negative cognitions are generally higher in people with EMSs irrespective of the situation. 

Keywords: early maladaptive schema, interpersonal relationships, interpersonal problems, interpersonal dysfunction, rejection, vignette, scenario, moderation, experiment” 

2. I think that Introduction provided a comprehensive setting of the context related to the issue addressed by the investigation. However, I found it to be quite lengthy for a journal research article, and seems more attuned to a thesis or dissertation. Some information within each subsection is redundant. I suggest for the authors to significantly reduce this section.

REPLY: The introduction section has been significantly reduced in size to reduce redundancy and limit article length. See track changes for further details. 

3. A similar observation with the subsection on the study aims. You may want to be explicit and succinct in reporting, instead of providing context.

REPLY: The study aims section has also been significantly reduced in size. See track changes for further details.

4. Include an ethics statement at the start of the methods section.

REPLY: The following statement has been added to the beginning of the methods section of the paper: “This research was conducted with approval from the University of New England Human Research Ethics Committee (Approval Number: HE18-248) with all participants providing informed and signed written consent for participation.”

5. The methods section is quite lengthy and may benefit from editing. There are times when you explain things, instead of just reporting explicitly. I suggest for the author to consult the STROBE checklist for the minimum information that should be reported in a correlation study such as this.

REPLY: The method section has been significantly reduced by deleting explanations for decisions and limit the section to explicit reporting. Repetition in text where information is already provided in appendices has also been used to reduce the methods length (see track changes for further details). 

6. Change the subsection title power analysis to reflect sample size instead of the statistical process.

REPLY: The subsection “Power analysis” has now been changed to “Sample size”

7. To clarify, your recruitment was not age group sensitive, hence older age participants who are enrolled in the 1st year psychology program were sampled? What was your inclusion criteria to begin with? Were there any race/nationality/ethnicity variances?

REPLY: The following statement was added to clarify the inclusion criteria “Any participant who was living in Australia, spoke English and was 18-years or older was allowed to participate in the present study.” Information about ethnicity was not collected as part of data collection and all demographic information related to participants was reported in the Participants subsection. 

8. Include description of scores and how are they typically interpreted for your instruments

REPLY: The following descriptions of scores and their interpretation were added for each questionnaire as follows:

YSQ-SF-3: “Mean scores for each EMS subscale range from one to six where scores between zero and three are considered of no clinical significance and scores greater than or equal to four are considered clinically significant (Shorey et al., 2015; Young & Brown, 2005). Scores across the 18 EMS scores can also be used to calculate five domain scores by averaging scores for EMSs that share common themes (e.g., rejection and disconnection). However, there are currently no guidelines or norms available for the interpretation of EMS domains or overall EMS scores.”

IIP-32: “The IIP-32 also provides T-scores based on a normative sample of 800 adults aged 18 to 89-years-old for clinical interpretation of scores across subscales.”

RSQ: “Overall rejection sensitivity score can be interpreted by comparing the deviation of the score from a mean rejection sensitivity score based on a sample of 685 healthy adults who completed the questionnaire electronically (M = 8.61, SD = 3.61; Berenson et al., 2009).”

DASS-21: “Individual scores within each subscale are added to produce an overall score which can fall between the “Normal” to “Extremely severe” range based on the deviation of scores from a normative sample of 1794 non-clinical adults (Henry & Crawford, 2005).”

9. Clarify which among the instruments were administered electronically. Did you at the very least compute for any form of psychometric properties of the electronic version of instrument you have used?

REPLY: The following section has been reworded to clarify which questionnaires were paper-based and which were electronically distributed “Participants were first sent a paper-based survey in the mail asking them to respond to some demographic questions and to complete the YSQ-SF-3 and IIP-32 due to the YSQ-SF-3 electronic distribution restrictions. Immediately after completing the paper-based survey, participants were directed to a Qualtrics (Version 2020, Qualtrics, Provo, UT) link to complete the online experiment and RSQ”.

Internal consistency was assessed for each questionnaire used in the present study with Cronbach’s alpha being comparable to those previously reported in research. Both Cronbach’s alpha from previous research and in the current study are reported under each questionnaire in the Measures subsection. This includes the RSQ which was the only questionnaire administered electronically which is also how the questionnaire was normed. This information is now provided in the methods section under “Measures”. 

10. I think you may need to have a separate subsection to fully describe the experimental paradigm (i.e., development, technical and electronic requirements, overall procedures).

REPLY: The experimental paradigm is now described under the subheading “Experiment Stimuli” which has been separated from information about its implementation which is now under the subheading “Procedure”.

11. Did you use the generic Jamovi software or did you include a specific statistical module? If you did the latter, do you have evidence to support the robustness of that module in carrying out the intended statistical tests?

REPLY: The following sentence was added to specify the specific statistical module along with references supporting its robustness: “The analyses were conducted in Jamovi (Version 1.6, Jamovi Project, 2021) using the GAMLj module for General Analyses of Linear Models (Version 2.6.6, Gallucci & Love, 2018) which is a robust module for estimating general linear, mixed and generalised multi-level models (Abbasnasab Sardareh et al., 2021; Titz, 2020).”

12. How different are the text results from those reported in the tables? Maybe highlight the most significant or interesting findings in the text, because as it is right now, this section 9as with the rest of the submitted manuscript) is quite text-heavy.

REPLY: The tables report the overall model specifications which are typically reported in the results for multi-level models. This includes the overall model specifications for each dependent variable including overall F tests, standardised regression coefficients, model R-squared, log-likelihood ratio tests and variance between and within participants. While these findings are important for the interpretation of the overall model and model building process they only provide the overall F-tests for the hypotheses and do not include pairwise comparisons. 

In contrast, the text results report pairwise comparisons between condition groups and analysis of simple slopes in addition the effect sizes related to each hypothesis. The only overlap between the tables and written results are the overall F-tests for the omnibus test of significance for rejection condition, EMS and condition*EMS which are repeated in text to justify proceeding with pairwise comparisons and providing an explanation of the observed effect in relation to the hypothesis. The effect size for the F-tests are also provided in the written results section but not in the tables as providing effect sizes for each standardised regression coefficient is redundant and not necessarily meaningful for the interpretation of the study hypotheses given the regression coefficients are insufficient for interpreting each pairwise comparison and do not provide information about the group means and variance. 

As a result, we have reduced text in the results section by referring readers to the tables for non-significant F-tests related to hypotheses and condensed information where there are multiple variables that had non-significant findings. 

We have also deleted results related to the statistical analysis of significant simple slopes as this information can be observed visually by inspection of Figures 2, 3 and 4 and the respective error bars. 

11. In your discussion, instead of using the hypotheses as subsection titles, you may want to rethink of a more descriptive title encompassing of what is discussed in the respective subsections.

REPLY: The hypotheses subtitles in the discussion section have now been change to “Interpersonal responses to rejection” and “The moderating effect of EMSs on interpersonal responses to rejection”. 

12. Were there any interesting or novel findings not found in the current evidence? Or how does your result differ among the studies your found?

REPLY: Unexpected findings presented in the discussion section have been expanded to discuss how findings differ to previous empirical evidence and expanded to provide an explanation for why this might have occurred (See track changes for more specific details. See below for paragraphs that have been expanded): 

1. Low intentional and global attributions following rejection:

“This finding opposes previous experimental research where participants reported increased perceptions of hostile intent following rejection in children (Reijntjes et al., 2011). Unlike the experiment conducted by Reijntjes et al. (2011) where rejection occurred from peers not known to the participants, in this study rejection was presented as a single transgression within a close interpersonal relationship (i.e., from a family member, friend or romantic partner). Fletcher and Kerr (2010) found that individuals were more likely to make positive attribution biases about a person’s actions and transgressions in relationships where there was greater intimacy. Therefore, if participants held relatively strong relationships with their loved ones, it is likely that they might attribute situations involving acceptance to be intentional and stable and unlikely that they would attribute the rejection to be stable or intentional following a single transgression. In contrast, it remains possible that in relationships with limited intimacy (e.g. strangers or peers unknown to the participant) participants might respond more negatively following rejection than how they responded in the present study (Reijntjes et al., 2011).”

2. Increased supporting behaviour in rejection condition

“This finding is surprising considering research generally suggests that people tend to show reduced pro-social behaviour following rejection (Twenge et al., 2001). However, people with low social anxiety have been found to engage in increased pro-social behaviour following rejection while people high in social anxiety did not suggesting some interpersonal characteristics might promote supportive behaviours following rejection (Weerdmeester & Lange, 2019).”

3. EMSs only moderating the relationship between rejection and select cognitions 

“While these findings provide support for the Schema Therapy model, which argues that individuals with EMSs show increased levels of distress and negative cognitions when EMSs are activated by stressful interpersonal contexts (e.g., rejection), it is surprising that EMSs only moderated the effect of rejection on self-blame and perceived rejection. Together these findings suggest that individuals with higher EMSs tend to report greater distress and negative cognitions regardless of the degree of the interpersonal context or situation. Given the finding that interpersonal rejection only activated select negative cognitions in people with higher EMSs, it is possible that EMSs might only moderate the relationship between interpersonal contexts and cognitions unique to that individual’s schema (e.g. only cognitions related to abandonment might be activated in people with abandonment schema). Further research could help clarify these findings by assessing the moderating effect of EMSs on cognitions and scenarios congruent to a specific EMS rather than EMSs in general.”

---

## [Decision Letter · Decision Letter 1]

15 Mar 2023

PONE-D-22-32865R1Assessing the relationship between early maladaptive schemas and interpersonal problems using interpersonal scenarios depicting rejectionPLOS ONE

Dear Dr. Janovsky,

Thank you for submitting your manuscript to PLOS ONE. After careful consideration, we feel that it has merit but does not fully meet PLOS ONE’s publication criteria as it currently stands. Therefore, we invite you to submit a revised version of the manuscript that addresses the points raised during the review process.

We look forward to receiving your revised manuscript.

Kind regards,

Delphine Grynberg, PhD

Academic Editor

PLOS ONE

Journal Requirements:

Reviewers' comments:

Reviewer's Responses to Questions

**Comments to the Author**

1. If the authors have adequately addressed your comments raised in a previous round of review and you feel that this manuscript is now acceptable for publication, you may indicate that here to bypass the “Comments to the Author” section, enter your conflict of interest statement in the “Confidential to Editor” section, and submit your "Accept" recommendation.

Reviewer #2: (No Response)

2. Is the manuscript technically sound, and do the data support the conclusions?

Reviewer #2: Partly

3. Has the statistical analysis been performed appropriately and rigorously? 

Reviewer #2: Yes

4. Have the authors made all data underlying the findings in their manuscript fully available?

Reviewer #2: Yes

5. Is the manuscript presented in an intelligible fashion and written in standard English?

Reviewer #2: No

6. Review Comments to the Author

Reviewer #2: I appreciate the effort put in the by the authors for succinctly addressing my previous comments. After reading the revised manuscript, I just have some minor comments, which I have doubt that the authors will be able to clarify. I may have overlooked this during the first review, but in your methods, you used mixed methods design. What was the rationale behind this and how does the qualitative methods address your hypothesis? How does your qualitative results triangulate with your quantitative results?

7. PLOS authors have the option to publish the peer review history of their article (what does this mean?). If published, this will include your full peer review and any attached files.

Reviewer #2: No

---

## [Author Response · Author response to Decision Letter 1]

27 Mar 2023

28-Mar-23

Regarding: PONE-D-22-32865R1

Assessing the relationship between early maladaptive schemas and interpersonal problems using interpersonal scenarios depicting rejection

Dear Professor Grynberg, 

Thank you for the opportunity to revise our manuscript and thank you to reviewer two for their input and feedback. Please find our replies to the academic editor and reviewer in the comments below. 

Sincerely, 

Thomas Janovsky

School of Psychology

University of New England

Armidale NSW 2351

Australia

 

Response to Academic Editor

Journal Requirements:

REPLY: The reference list has been reviewed to ensure it is correct (see track changes). New references cited have now been added to the reference list. 

 

Response to Reviewers

Reviewer #2: 

I appreciate the effort put in the by the authors for succinctly addressing my previous comments. After reading the revised manuscript, I just have some minor comments, which I have doubt that the authors will be able to clarify. I may have overlooked this during the first review, but in your methods, you used mixed methods design. What was the rationale behind this and how does the qualitative methods address your hypothesis? 

REPLY: The study hypotheses related to assessing the cognitive, emotional and behavioural responses to interpersonal scenarios depicting acceptance, ambiguous rejection and rejection. When measuring cognitions and emotions, questions were identical across all scenarios to ensure that variations in questions between scenarios did not impact the validity of the results. In these questions, participants were able to rate the likelihood of cognitions and distress resulting in quantitative data we could analyse statistically to test the study hypotheses. 

However, behavioural responses to interpersonal scenarios were more challenging to assess quantitatively. Firstly, unlike cognitive and emotional responses, behavioural responses by nature depended on the external content of the scenarios which changed across control, ambiguous and rejection conditions. For example, a generic behavioural response such as “avoid them” would be confusing for some control scenarios (e.g. you text your friend to suggest an idea for catching up. They text you back suggesting a date and time. What would you do in response to this scenario?). Secondly, Weiner’s (1985) attribution theory which provided the basis for measuring cognitive responses is a well-established theory in psychology as is the use of subjective units of distress rating scales in cognitive behaviour therapy and research (e.g. Davidson et al., 2004; Woods et al., 2002). However, there are many different theories related to interpersonal behavioural – none of which were clearly superior to others for structuring questions that could be measured quantitatively for the purpose of the experiment (e.g. Interpersonal circumplex, schema coping styles). For example, Young et al.’s (2003) theory of coping styles are defined as intra-psychic responses (not necessarily behavioural responses) to a person’s early maladaptive schemas (Arntz et al., 2021; Young et al., 2003). Behavioural response constructs in many theories such as over-compensation (Schema coping styles: Young et al., 2003) and submissiveness (Interpersonal circumplex: Locke & Adamic, 2012) are also defined by whether the person reacts in a way disproportionate to the interpersonal context which in our experiment changed across conditions. Thirdly, researchers have argued that what is considered a maladaptive behavioural response varies based on context as behaviours that are maladaptive in some contexts can be adaptive in others (Fischer et al., 2021; Sih et al., 2004). For example, Fischer et al (2021) found that accepting and not acting to try to change a situation was only adaptive when the situation was unchangeable. 

Given the above variability in behavioural responses due to different theories of interpersonal behaviour and changing interpersonal context and ambiguity about what behaviours can be considered “unhelpful” in changing contexts, we used an embedded mixed methods design where open ended qualitative behavioural responses were measured within a predominantly quantitative study. The qualitative behavioural responses were then converted into quantitative variables for analysis and assessment of the hypotheses investigating whether (1) participants report changing behavioural responses as the degree of rejection increases across interpersonal scenarios and (2) whether EMSs moderated the relationship between interpersonal rejection and behavioural responses. 

To help clarify this design and rationale, the following paragraph was expanded and revised in the Experiment Stimuli section of the Methods: 

“Emotional distress associated with each scenario was measured by using a subjective units of distress scale asking participants to rate the degree to which they would be angry or upset following the scenario on a sliding scale from 0 (Not upset at all) to 100 (Extremely upset or angry). However, unlike attributions and subjective measures of distress, behavioural responses in grounded theory are often defined by how disproportionate they are to the individual context (e.g. the schema coping style of overcompensation) which meant questions related to specific behaviours based on theory would have needed to vary across scenarios making it difficult to differentiate whether changing responses related to the changes in condition or changes in behavioural response questions. Due to the possible variability of behavioural responses across contexts, behavioural responses were measured qualitatively embedded within a predominantly quantitative design. Behavioural responses were measured by asking participants to record a written response summarising how they would likely respond to the interpersonal situation and why. The behavioural responses were then coded using inductive content analysis to create a series of binary variables scoring endorsement or non-endorsement of each behavioural response so that behaviours could be assessed by statistical analysis with respect to the study hypotheses.”

References

Arntz, A., Rijkeboer, M., Chan, E., Fassbinder, E., Karaosmanoglu, A., Lee, C. W., & Panzeri, M. (2021). Towards a reformulated theory underlying Schema Therapy: Position paper of an international workgroup. Cognitive Therapy and Research, 45(6), 1007–1020. https://doi.org/10.1007/s10608-021-10209-5

Davidson, J.R.T., Foa, E.B., Huppert, J.D., Keefe, F.J., Franklin, M.E., Compton, J.S., Zhao, N., Connor, K.M., Lynch, T.R., & Gadde, K.M. (2004). Fluoxetine, comprehensive cognitive behavioral therapy, and placebo in generalized social phobia. Archives of General Psychiatry, 61(10), 1005-1013. https://doi.org/10.1001/archpsyc.61.10.1005.

Fischer, R., Scheunemann, J., & Moritz, S. (2021). Coping strategies and subjective well-being: Context matters. Journal of Happiness Studies, 22, 3413-3434.

https://doi.org/10.1007/s10902-021-00372-7

Locke, K. D., & Adamic, E. J. (2012). Interpersonal circumplex vector length and interpersonal decision making. Personality and Individual Differences, 53(6), 764–769. https://doi.org/10.1016/j.paid.2012.06.001

Sih, A., Bell, A.M., Johnson, J.C., & Ziemba, R.E. (2004). Behavioral syndromes: An integrative overview. The Quarterly Review of Biology, 79(3), 241-277. http://doi.org/10.1086/422893.

Weiner, B. (1985). An attributional theory of achievement motivation and emotion. Psychological Review, 92(4), 548–573. https://doi.org/10.1037/0033-295X.92.4.548

Woods, C.M., Chambless, D.L., & Skeketee, G. (2002). Homework compliance and behaviour therapy outcome for panic with agoraphobia and obsessive compulsive disorder. Cognitive Behaviour Therapy, 31(2), 88-95. https://doi.org/10.1080/16506070252959526

---

## [Decision Letter · Decision Letter 2]

19 Jun 2023

PONE-D-22-32865R2Assessing the relationship between early maladaptive schemas and interpersonal problems using interpersonal scenarios depicting rejectionPLOS ONE

Dear Dr. Janovsky,

Thank you for submitting your manuscript to PLOS ONE. After careful consideration, we feel that it has merit but does not fully meet PLOS ONE’s publication criteria as it currently stands. Therefore, we invite you to submit a revised version of the manuscript that addresses the points raised during the review process.

We look forward to receiving your revised manuscript.

Kind regards,

Delphine Grynberg, PhD

Academic Editor

PLOS ONE

Journal Requirements:

Additional Editor Comments:

As I could not find reviewers I reviewed it myself. I have only two minor comments1. I would add a sentence in the background section of your abstract to present the objective, rather than mentioning it in the method section

2. Would it be possible to better argue the importance to differentiate ambiguous vs clear rejection conditions? Did you expect increased distress in these type of situations and if yes, why? 

Reviewers' comments:

Reviewer's Responses to Questions

**Comments to the Author**

1. If the authors have adequately addressed your comments raised in a previous round of review and you feel that this manuscript is now acceptable for publication, you may indicate that here to bypass the “Comments to the Author” section, enter your conflict of interest statement in the “Confidential to Editor” section, and submit your "Accept" recommendation.

Reviewer #2: (No Response)

2. Is the manuscript technically sound, and do the data support the conclusions?

Reviewer #2: Yes

3. Has the statistical analysis been performed appropriately and rigorously? 

Reviewer #2: Yes

4. Have the authors made all data underlying the findings in their manuscript fully available?

Reviewer #2: Yes

5. Is the manuscript presented in an intelligible fashion and written in standard English?

Reviewer #2: Yes

6. Review Comments to the Author

Reviewer #2: (No Response)

7. PLOS authors have the option to publish the peer review history of their article (what does this mean?). If published, this will include your full peer review and any attached files.

Reviewer #2: No

---

## [Author Response · Author response to Decision Letter 2]

27 Jun 2023

26-Jun-23

Regarding: PONE-D-22-32865R1

Assessing the relationship between early maladaptive schemas and interpersonal problems using interpersonal scenarios depicting rejection

Dear Professor Grynberg, 

Thank you for the opportunity to revise our manuscript and for your input and feedback. Please find our replies in the comments below. 

Sincerely, 

Thomas Janovsky

School of Psychology

University of New England

Armidale NSW 2351

Australia

 

Response to Academic Editor

Journal Requirements:

REPLY: The reference list has been reviewed to ensure it is correct (see track changes). New references cited have now been added to the reference list. 

Reviewer #3: 

1. I would add a sentence in the background section of your abstract to present the objective, rather than mentioning it in the method section

REPLY: The study aims have now been mentioned in the background section rather than the methods section in the abstract:

“Background: Early maladaptive schemas (EMSs) have been theorised to contribute to reoccurring interpersonal problems. This study developed a novel experimental paradigm that aimed to assess if EMSs moderate the impact of interpersonal situations on interpersonal responses by manipulating the degree of rejection in a series of interpersonal vignettes depicting acceptance, ambiguous rejection and rejection. Method: In a sample of 158 first-year psychology students (27.2% male; 72.2% female; 0.6% other) participant responses to interpersonal scenarios were measured including degree of perceived rejection, emotional distress, conviction in varying cognitive appraisals consistent with attribution theory and behavioural responses to scenarios. Qualitative data was analysed using inductive content analysis and statistical analyses were conducted using multi-level mixed effect linear and logistic regression models using the software Jamovi (Version 1.6, Jamovi Project, 2021). Results: People reporting higher EMSs reported increased emotional distress (F(1, 156) = 24.85, p < .001), perceptions of rejection (F(1, 156) = 34.33, p < .001), self-blame (F(1, 156) = 53.25, p < .001), other-blame (F(1, 156) = 13.16, p < .001) and more intentional (F(1, 156) = 9.24, p = .003), stable (F(1, 156) = 25.22, p < .001) and global (F(1, 156) = 19.55, p < .001) attributions but no differences in reported behavioural responses. The results also supported that EMSs moderate the relationship between interpersonal rejection and perceptions of rejection (F(2, 1252) = 18.43, p < .001), emotional distress (F(2, 1252) = 12.64, p < .001) and self-blame (F(2, 1252) = 14.00, p < .001). Conclusion: Together these findings suggest that people with EMSs experience increased distress and select negative cognitions in situations where there are higher levels of rejection but that distress and negative cognitions are generally higher in people with EMSs irrespective of the situation.”

2. Would it be possible to better argue the importance to differentiate ambiguous vs clear rejection conditions? Did you expect increased distress in these type of situations and if yes, why?

REPLY: The following information has been added to the introduction section to help clarify the importance of differentiating ambiguous versus clear rejection and what might be expected/why based on previous study findings:

“The importance given to interpersonal contexts is argued by interpersonal theorists who have found that interpersonal behaviour is open to change across different contexts such as increased interpersonal pressures or responses from others (Locke & Adamic, 2012). For example, Farc et al. (2008) found that people who experienced physical abuse during childhood were more likely to perceive aggression in children when the cues in the environment were ambiguous. Similar findings have been found in other studies showing increased endorsement of hostility schemas to be associated with increased perceptions of hostile intentions in others when information about another person’s behaviour was ambiguous (Kim et al., 2021). Furthermore, people with higher rejection expectations reported increased withdrawal from a group when rejection in an interpersonal scenario was unambiguous compared to ambiguous (Nesdale et al., 2014; Zimmer-Gembeck & Nesdale, 2012).Together these findings suggest that individuals with EMSs are more likely to report negative responses (e.g. negative cognitions and behavioural responses) to interpersonal contexts depicting rejection and that responses are likely to become increasingly negative as the ambiguity about rejection in the scenario decreases. These findings are consistent with theoretical models depicting schemas (e.g. Schema therapy model, the rejection sensitivity model) arguing that people with stronger negative beliefs or schemas of themselves and others are more likely to perceive rejection in the environment and that perceptions of rejection are associated with higher levels of distress and more negative interpersonal behaviour responses (Young et al., 2003; Zimmer-Gembeck & Nesdale, 2012).”

The following sentence was also added to the Aims and Hypotheses section of the paper to re-iterate the above information with respect to the current study design: 

“Measuring degree of rejection based on acceptance, ambiguous and unambiguous rejection conditions is consistent with previous studies investigating cognitive, emotional and behavioural responses in people with negative relationship expectancies such as rejection sensitivity and hostility schema (e.g. Kim et al., 2021; Zimmer-Gembeck & Nesdale, 2012).”

---

## [Editor Report · Decision Letter 3]

29 Jun 2023

Assessing the relationship between early maladaptive schemas and interpersonal problems using interpersonal scenarios depicting rejection

PONE-D-22-32865R3

Dear Dr. Janovsky,

We’re pleased to inform you that your manuscript has been judged scientifically suitable for publication and will be formally accepted for publication once it meets all outstanding technical requirements.

Kind regards,

Delphine Grynberg, PhD

Academic Editor

PLOS ONE
---

## [Editor Report · Acceptance letter]

6 Jul 2023

PONE-D-22-32865R3 

Assessing the relationship between early maladaptive schemas and interpersonal problems using interpersonal scenarios depicting rejection 

Dear Dr. Janovsky:

I'm pleased to inform you that your manuscript has been deemed suitable for publication in PLOS ONE. Congratulations! Your manuscript is now with our production department. 

Kind regards, 

on behalf of

Dr. Delphine Grynberg 

Academic Editor

PLOS ONE